# CD8+ T-cell responses towards conserved influenza B virus epitopes across anatomical sites and age

Tejas Menon [1,10], Patricia T. Illing [2,10], Priyanka Chaurasia[2,10], Hayley A. McQuilten [1], Chloe Shepherd[2], Louise C. Rowntree [1], Jan Petersen[2], Dene R. Littler [2], Grace Khuu[2], Ziyi Huang[2], Lilith F. Allen[1], Steve Rockman[1,3], Jane Crowe[4], Katie L. Flanagan[5,6,7], Linda M. Wakim[1], Thi H. O. Nguyen [1], Nicole A. Mifsud [2], Jamie Rossjohn [2,8,11], Anthony W. Purcell [2,11], Carolien E. van de Sandt [1,9,11] & Katherine Kedzierska [1,11] ✉

Influenza B viruses (IBVs) cause substantive morbidity and mortality, and yet immunity towards IBVs remains understudied. CD8+ T-cells provide broadly cross-reactive immunity and alleviate disease severity by recognizing conserved epitopes. Despite the IBV burden, only 18 IBV-specific T-cell epitopes restricted by 5 HLAs have been identified currently. A broader array of conserved IBV T-cell epitopes is needed to develop effective cross-reactive T-cell based IBV vaccines. Here we identify 9 highly conserved IBV CD8+ T-cell epitopes restricted to HLA-B*07:02, HLA-B*08:01 and HLA-B*35:01. Memory IBV-specific tetramer+CD8+ T-cells are present within blood and tissues. Frequencies of IBV-specific CD8+ T-cells decline with age, but maintain a central memory phenotype. HLA-B*07:02 and HLA-B*08:01-restricted NP$_{30-38}$ epitope-specific T-cells have distinct T-cell receptor repertoires. We provide structural basis for the IBV HLA-B*07:02-restricted NS1$_{196-206}$ (11-mer) and HLA-B*07:02-restricted NP$_{30-38}$ epitope presentation. Our study increases the number of IBV CD8+ T-cell epitopes, and defines IBV-specific CD8+ T-cells at cellular and molecular levels, across tissues and age.

Co-circulating seasonal influenza A (IAV) and influenza B (IBV) viruses cause annual epidemics, resulting in 3–5 million severe clinical infections and 290,000–650,000 fatal cases annually[1]. IBVs, consisting of B/Victoria and B/Yamagata lineages[2], typically make up 25% of annual influenza cases worldwide, although in some seasons they can exceed 50% of all influenza cases[3,4]. The burden of seasonal IBV infections is greatly underappreciated[2]. Hospitalization rates of seasonal IBVs are second only to seasonal A/H3N2 viruses and higher than that of seasonal A/H1N1 viruses[5]. In contrast to IAVs, which significantly impact young children (<2 years old) and the elderly (>65 years)[6], the burden

[1]Department of Microbiology and Immunology, University of Melbourne, at the Peter Doherty Institute for Infection and Immunity, Parkville, VIC, Australia. [2]Infection and Immunity Program & Department of Biochemistry and Molecular Biology, Biomedicine Discovery Institute, Monash University, Clayton, VIC, Australia. [3]CSL Seqirus Ltd, Parkville, VIC, Australia. [4]Deepdene Surgery, Deepdene, VIC, Australia. [5]Tasmanian Vaccine Trial Centre, Launceston General Hospital, Launceston, TAS, Australia. [6]School of Health Sciences and School of Medicine, University of Tasmania, Launceston, TAS, Australia. [7]School of Health and Biomedical Science, RMIT University, Melbourne, VIC, Australia. [8]Institute of Infection and Immunity, Cardiff University School of Medicine, Cardiff, UK. [9]Department of Hematopoiesis, Sanquin Research and Landsteiner Laboratory, Amsterdam UMC, University of Amsterdam, Amsterdam, The Netherlands. [10]These authors contributed equally: Tejas Menon, Patricia T. Illing, Priyanka Chaurasia. [11]These authors jointly supervised this work: Jamie Rossjohn, Anthony W. Purcell, Carolien E. van de Sandt, Katherine Kedzierska. ✉e-mail: kkedz@unimelb.edu.au

of IBV infections is especially high among school-aged children (5–13 years) in terms of incidence[3,7], hospitalization[8,9] and fatal outcomes[10,11]. During the COVID-19 pandemic, non-pharmacological interventions in response to SARS-CoV-2 significantly reduced transmission of IAV and IBV to negligible levels worldwide[12]. Most notably, influenza B viruses from the Yamagata lineage have not been detected since early 2020[12]. However, with easing of COVID-19 restrictions, seasonal influenza viruses have re-emerged[12], leading to concerns for severe influenza seasons in the near future as a result of reduced population immunity[13]. This is further aggravated by potential co-infections of influenza and SARS-CoV-2[14] or large amounts of single infections with either influenza or SARS-CoV-2[15]. As such, public health measures, including vaccinations, remain essential in curbing the re-emerging threats.

Current seasonal inactivated influenza vaccines rely on strain-specific antibodies that target the hemagglutinin (HA) and neuraminidase (NA) proteins on the surface of influenza viruses[16,17]. However, antigenic drift of HA and NA proteins results in reduced efficacy of these vaccines, requiring near annual updates of the influenza vaccine strains[18]. In the absence of neutralizing antibodies, cross-protective CD8[+] T cells can provide substantial protection against severe influenza disease[19,20] by recognizing often highly conserved influenza peptides presented on human leukocyte antigen class I molecules (HLA-I) on the surface of virus-infected cells. As such, new T cell based influenza vaccine approaches aim to fully utilize the potential of cross-protective CD8[+] T cells[16]. Indeed novel mRNA vaccine platforms have successfully induced influenza-specific CD8[+] T cell responses in mice and ferrets[21–23].

While IBVs should clearly not be underestimated, the lack of a zoonotic reservoir limits their pandemic potential, which means that they are less well studied in comparison to IAVs. However, the lack of zoonotic reservoirs also means that IBV infections and transmissions can potentially be well controlled with a rationally designed broadly cross-reactive IBV vaccine targeting T cells. As immunity to IBVs is understudied, only 18 IBV CD8[+] T cell epitopes restricted by 5 HLAs have been defined to date[24–29], including 14 IBV epitopes identified by our group, 5 of which are restricted by the highly prevalent HLA-A*02:01 allomorph[24,27] and 6 restricted by HLA-A*24:02, prevalent among indigenous and Asian populations[26]. This is in stark contrast to IAVs, for which >400 CD8[+] T cell epitopes have been identified[30]. The lack of known IBV epitopes not only hampers the development of universal IBV T cell vaccines, but also greatly impairs our ability to study protective versus detrimental IBV-specific immune responses across the human lifespan.

An important consideration for designing future CD8[+] T cell-based influenza vaccines is maximizing global HLA coverage to ensure broad population coverage. In our study, we utilized immunopeptidomics[24–26,31] to identify IBV peptides presented by HLA-B*07:02, HLA-B*08:01 and HLA-B*35:01 allomorphs. We identified 9 IBV CD8[+] T cell epitopes restricted by prominent HLA-B*07:02 (B7/NP$_{30-38}$, B7/NP$_{233-244}$, B7/NS1$_{196-206}$, B7/HA$_{255-263}$), HLA-B*08:01 (B8/NP$_{92-99}$, B8/NP$_{479-487}$) and HLA-B*35:01 (B35/NS1$_{260-268}$, B35/HA$_{231-239}$, B35/PB$_{66-74}$) alleles. Most T cell epitopes were highly conserved among known influenza B virus strains (>99%). Our study thus increased the overall number of known IBV CD8[+] T cell epitopes by 50%, presented by 2 additional prominent HLAs, HLA-B*07:02 and HLA-B*35:01. Additionally, we performed in-depth ex vivo analyses for 6 most prominent IBV-specific CD8[+] T cell populations in human peripheral blood mononuclear cells (PBMCs) across human tissue compartments and across the human lifespan.

## Results

### Global prevalence of HLA-B*07:02, HLA-B*08:01 and HLA-B*35:01 allomorphs

As only 18 CD8[+] T cell epitopes spanning 5 HLAs have been previously described for clinically relevant and understudied IBVs[24–29], we sought to identify IBVs derived CD8[+] T cell epitopes and presented by highly prevalent HLA-B*07:02, HLA-B*08:01 and HLA-B*35:01 allotypes (source: http://www.allelefrequencies.net/ accessed 3/03/2023). HLA-B*07:02 is found in 5.5% of the global population and is particularly enriched in European and North-East Asian populations (9.1% and 6.0%, respectively) (Fig. 1a). HLA-B*08:01 is present in 4.4% of the global population, with highest prevalence in Europeans (7.5%), followed by Australians (5.7%) and North African populations (5.2%) (Fig. 1a). HLA-B*35:01 is found in 5.9% of the global population, with highest prevalence in North-Americas (9.7%), followed by South and Central American populations (7.3%) and North-East Asian (7.1%) (Fig. 1a).

In our cohort of healthy Australians, all three HLA allomorphs of interest were broadly co-expressed with other HLA-A, B and C allomorphs. HLA-B*07:02 and B*08:01 were co-expressed in 14–18% of donors, while all three HLA allomorphs were highly co-expressed with HLA-B*44:02 (HLA-B*07:02, 16.3%; HLA-B*08:01, 12.5%; HLA-B*35:01, 16.0%) (Fig. 1b). Commonly shared co-expression allomorphs were detected for HLA-A and -C, namely HLA-A*01:01, A*02:01, A*03:01, C*07:01 and C*07:02, reflective of their prominent expression in the global population (Fig. 1c).

Based on the global frequencies and co-expressions, we concluded that any potential identified epitopes restricted to our three HLA alleles of interest would significantly increase the population coverage when included in potential T cell-based influenza vaccine strategies.

### CD8[+] T cell responses towards known HLA-B*08:01-restricted IBV epitopes

The 18 previously identified IBV CD8[+] T cell epitopes are restricted to HLA-A*02:01[24,28], HLA-A*03:01[29], HLA-A*11:01[25], HLA-A*24:02[26] or HLA-B*08:01[27] (Fig. 1d). To the best of our knowledge, no IBV CD8[+] T cell epitopes have been reported for HLA-B*07:02 and HLA-B*35:01, while 3 HLA-B*08:01-restricted IBV epitopes, B8/NP$_{30-38}$ (RPIIRPATL), B8/NP$_{263-271}$ (ADRGLLRDI) and B8/NP$_{413-422}$ (ALKCKGFHV), were identified by incubating peripheral blood mononuclear cells (PBMCs) from one donor with IBV, then screening peptides in a $^{51}$Cr cytotoxicity assay[27]. To understand the immunogenicity of the candidate HLA-B*08:01-restricted CD8[+] T cell epitopes[27], we probed PBMCs from six healthy HLA-B*08:01-expressing individuals for IBV-specific CD8[+] T cell responses using our in vitro peptide expansion approach combined with intracellular cytokine staining (ICS) for IFN-γ/TNF production[24–26,32] (Fig. 1e and Supplementary Fig. 1). Robust IFN-γ[+]TNF[+] CD8[+] T cell responses were observed for PBMCs stimulated with the NP$_{30-38}$ peptide across all HLA-B*08:01 donors (median of 2.2%), while no responses were detected for NP$_{263-271}$ and NP$_{413-422}$ (Fig. 1e), consistent with findings by others[20].

Overall, there remains a paucity of data on IBV-specific CD8[+] T cell epitopes across predominant class I HLAs.

### Identification of IBV-derived peptides presented by HLA-B*07:02, -B*08:01 or -B*35:01

To identify IBV peptides presented by HLA-B*07:02, HLA-B*08:01 or HLA-B*35:01, we used our well-established immunopeptidomics methodology[24–26]. HLA-B*07:02-, HLA-B*08:01- or HLA-B*35:01-expressing C1R cell lines were infected with IBV (B/Malaysia/2506/04) for 12 h followed by HLA-I isolation using the pan HLA-I antibody W6/32 and analyzed by liquid chromatography-tandem mass spectrometry (LC-MS/MS). Human proteome derived peptides identified at a 1% false discovery rate (FDR) were consistent in length with HLA-I ligands. All datasets were dominated by 9-mer peptides (Fig. 2a), with HLA-B*08:01 also showing a high prevalence of 8-mer peptides which has been reported before[33]. HLA-B*07:02 and HLA-B*35:01 shared a preference for Proline at P2, while at PΩ (the C-terminal amino acid), B*07:02 showed a hierarchy of Leu»Phe/Val/Met/Ile and B*35:01 of

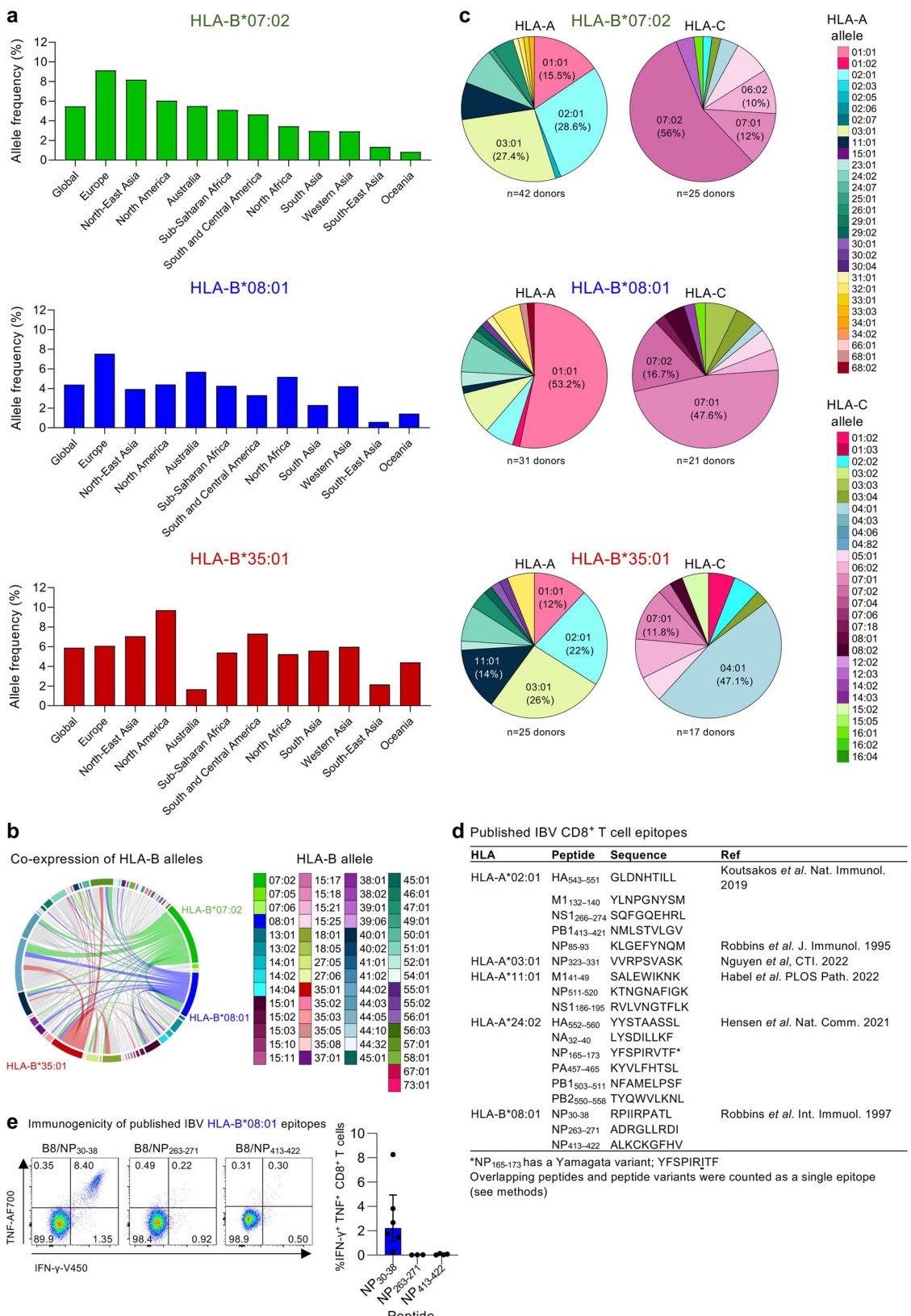

**Fig. 1 | Global expression of HLA-B\*07:02, HLA-B\*08:01 and HLA-B\*35:01 alleles and characterization of published IBV epitopes. a** Allele frequency of HLA-B\*07:02, HLA-B\*08:01 and HLA-B\*35:01 according to geographical location (source: http://www.allelefrequencies.net/ accessed 3/03/2023). **b** Co-expression of HLA-B alleles. **c** Analysis was performed using our database of healthy blood donor buffy pack (n = 185 donors). **d** Published IBV CD8+ T cell epitopes. **e** Representative FACS plots of IFN-γ and TNF expression, together with the frequency of IFN-γ+TNF+CD8+ T cells following in vitro stimulation with previously identified IBV HLA-B\*08:01-restricted peptides (n = 6 donors for $NP_{30-38}$, n = 3 donors for $NP_{263-271}$ and n = 4 donors for $NP_{413-422}$). DMSO background was subtracted. Bars represent median and interquartile range (IQR). Source data are provided as a Source Data file.

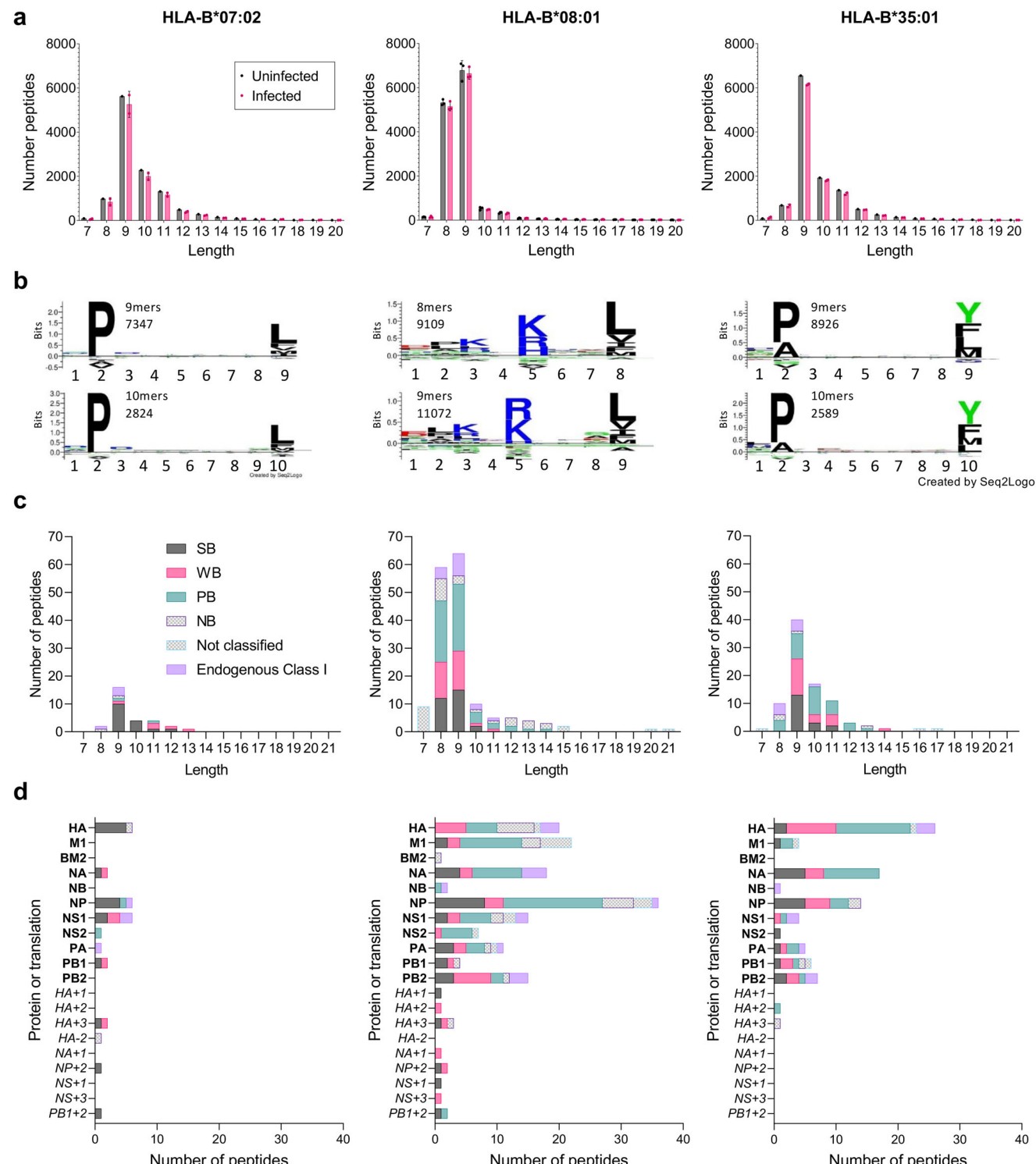

Tyr>Leu/Phe/Met (Fig. 2b). In contrast, HLA-B*08:01 showed enrichment of positively charged amino acids (Arg and Lys) at P5 and, to a lesser extent, P3 of both 8- and 9-mer peptides, with Leu dominating PΩ. All of these anchor residues are consistent with previous findings[34].

To identify IBV peptides, a 5% peptide FDR was employed to minimize exclusion of flu derived peptides, with the data from uninfected cells utilized as an additional filter for false discovery[24-26]. IBV peptide sequences identified in isolations from HLA class II of C1R were excluded, however all other peptides were retained in the analysis and their predicted binding to both the transfected and endogenous HLA-I calculated with NetMHC4.0 (Supplementary

Data 1). Twenty-four, 114, and 71 IBV peptide sequences were identified as binders of HLA-B*07:02, HLA-B*08:01 and HLA-B*35:01, respectively (Fig. 2c). For all three transfectant HLA-I, the length distribution of IBV peptides mirrored that observed for human peptides, dominated by 9-mers, with the HLA-B*08:01 data set also containing a large proportion of 8-mer peptides (Fig. 2c). Broad sampling of the IBV proteome was observed for HLA-B*08:01 and HLA-B*35:01, dominated by NP, NA, M1, HA, PB2, and NS1 for HLA-B*08:01, and HA, NA, and NP for HLA-B*35:01 (Fig. 2d). HLA-B*07:02 sampled NP, HA, NS1, PB1, NA and NS2. We also identified potential peptide binders from alternative reading frames for

**Fig. 2 | Identification of IBV peptides restricted by HLA-B*07:02, HLA-B*08:01 and HLA-B*35:01. a** Length distribution of human proteome-derived peptides (non-redundant by sequence) identified at an FDR of 1% from immunopurification of HLA class I (using the pan HLA-I antibody W6/32) of C1R.B*07:02, C1R.B*08:01 and C1R.B*35:01. All data sets were filtered for peptides identified from the endogenous HLA-C*04:01 and HLA class II of C1R cells from previous experiments[24,26], and peptides identified as possessing the HLA-C*04:01 binding motif via Gibbs cluster analysis. The C1R.B*08:01 data sets were also filtered of peptides isolated using W6/32 from the endogenous HLA-I of C1R. Human proteome derived peptides from uninfected and infected replicates are shown in black and pink, respectively. Points represent individual replicates, bars represent the mean number of peptides, error bars are standard deviation. Numbers of replicates: C1R.B*07:02−1 uninfected, 2 B/Malaysia infected; C1R.B*08:01−3 uninfected, 3 B/Malaysia infected; C1R.B*35:01−1 uninfected, 2 B/Malaysia infected. **b** Sequence motifs of preferred lengths for HLA-B*07:02 (9 and 10), HLA-B*08:01 (8 and 9) and HLA-B*35:01 (9 and 10). Motifs are based on human peptides of these lengths represented in (**a**). Motifs were made using Seq2Logo. **c** Length and (**d**) proteome (and alternative translation frame product; italicized) distribution of influenza B-derived peptides isolated in experiments using C1R.B*07:02, C1R.B*08:01 or C1R.B*35:01. Peptides predicted to bind the overexpressed HLA are denoted as strong binders (SB), weak binders (WB) or possible binders (PB) as per Supplementary Data 1. Peptides isolated in experiments that were not predicted to bind the overexpressed HLA are separated into the following categories: non-binder (NB)−8−14 amino acids in length and not predicted to bind any HLA expressed, Endogenous Class I−predicted to bind endogenous HLA class I of C1R as described in Supplementary Data 1, not classified−<8 or >14 amino acids length therefore binding prediction not performed. Source data are provided as a Source Data file.

all three HLAs, including NP + 2$_{180-188}$ and PB1 + 2$_{80-89}$, which were common to HLA-B*07:02 and HLA-B*08:01.

The three previously identified HLA-B*08:01-restricted IBV peptides (NP$_{30-38}$, NP$_{263-271}$ and NP$_{413-422}$) were all conserved in B/Malaysia/2506/04, however, only NP$_{30-38}$ (RPIIRPATL) was identified in our immunopeptidome data, consistent with the observed lack of CD8$^+$ T cell response towards NP$_{263-271}$ and NP$_{413-421}$ (Fig. 1e)[20]. NP$_{30-38}$ was also one of eight peptides identified as a ligand for both HLA-B*07:02 and -B*08:01, despite their disparate binding motifs (Supplementary Data 1). Furthermore, NP$_{30-38}$ was also not an HLA-B*35:01 ligand despite having anchor residues consistent with other -B*35:01 ligands (Fig. 2b).

A selection of 21 HLA-B*07:02 peptides and 55 HLA-B*35:01 peptides were chosen for immunogenicity screening in vitro based on a combination of high quality of peptide identification and binding prediction for the transfected HLA-I. NP$_{30-38}$ was also included as a potential HLA-B*35:01 peptide due to its anchor residues. For HLA-B*08:01, selections were further focused on in frame proteins, with 35 HLA-B*08:01 peptides selected for screening, including NS2$_{11-19}$ due to strong binding prediction despite identification at >5% FDR (Supplementary Data 1).

### Identification of immunogenic IBV-specific HLA-B*07:02-, B*08:01- and HLA-B*35:01-restricted IBV peptides

To define CD8$^+$ T cell immunogenicity towards our LC-MS/MS-identified IBV-derived peptides, we assigned those 112 peptides into 7 HLA specific peptide pools based on increasing predicted binding affinity towards their respective HLA (Supplementary Table 1). PBMCs obtained from healthy HLA-B*07:02-, HLA-B*08:01- or HLA-B*35:01-expressing individuals (Supplementary Table 2) were stimulated with their respective peptide pools for 10–12 days in vitro, followed by restimulation with the respective pool to measure IFN-γ and TNF production (Fig. 3a). CD8$^+$ T cells responding to specific pools were further dissected using individual IBV-derived peptides.

For HLA-B*07:02, both pool A and B stimulated IFN-γ and TNF production (Fig. 3b). Dissection of pool A and B identified five immunogenic HLA-B*07:02 peptides; NP$_{30-38}$, NP$_{233-244}$, NS1$_{196-206}$, NS2$_{62-70}$ and HA$_{255-263}$ responses in 5/6, 5/6, 3/4, 2/4, 2/4 donors respectively. For HLA-B*08:01, pool C stimulated IFN-γ and TNF production but not pool D (Fig. 3c and Supplementary Fig. 2a). Dissection of pool C revealed four immunogenic HLA-B*08:01 peptides; NP$_{30-38}$ and NP$_{92-99}$ responses in 5/6 and 3/6 donors respectively, whereas NP$_{479-487}$ and NP$_{468-476}$ induced a robust IFN-γ$^+$TNF$^+$CD8$^+$ T cell response in one individual. HLA-B*35:01-derived pools E and F stimulated IFN-γ and TNF production, but not pool G (Fig. 3d and Supplementary Fig. 2a). Dissection of pools E and F identified five immunogenic HLA-B*35:01 peptides; NS1$_{260-268}$, HA$_{231-239}$, PB2$_{66-70}$ NP$_{165-173}$ and NP$_{468-476}$ responses in 4/6, 3/6, 4/6, 2/6 and 1/6 donors, respectively. Two donors responding to NP$_{165-173}$ also expressed HLA-A*24:02.

Overall, we identified 13 immunogenic IBV peptides in HLA-B*07:02- (5 peptides), HLA-B*08:01- (3 peptides) and HLA-B*35:01- (5 peptides) expressing individuals. The NP$_{30-38}$ peptide was of particular interest as it was immunogenic in both HLA-B*07:02 and/or HLA-B*08:01-expressing donors, but no responses were observed for HLA-B*35:01-expressing donors (pool G; Fig. 3d).

### Verification of HLA restriction for the immunogenic IBV-derived peptides

Following identification of IBV-derived immunogenic peptides in HLA-B*07:02-, HLA-B*08:01- and HLA-B*35:01-expressing individuals, we subsequently verified the HLA restriction for the identified IBV peptides to rule out any potential peptide presentation by unrelated HLA alleles co-expressed in our donors. Particularly the B*35:01 associated NP$_{165-173}$ peptide only generated responses in HLA-A*24:02$^+$ and HLA-B*35:01$^+$ individuals and was previously identified as an HLA-A*24:02-restricted epitope by our group[26]. We generated three new peptide pools consisting of 5 immunogenic HLA-B*07:02 peptides (pool H), 4 immunogenic HLA-B*08:01 peptides (pool I) and 5 immunogenic HLA-B*35:01 peptides (pool J) (Supplementary Table 3), for stimulating CD8$^+$ T cells. Pool-specific CD8$^+$ T cells were restimulated with C1R cell lines expressing HLA-B*07:02, HLA-B*08:01 or HLA-B*35:01 molecules which were pulsed with individual peptides from the corresponding pool and stained for IFN-γ/TNF production (Fig. 4a). Pool-specific CD8$^+$ T cells were also restimulated with peptide-pool pulsed parental C1Rs to control for the low expression of HLA-B*35:03 and HLA-C*04:01 by C1Rs, which may elicit CD8$^+$ T cell responses specific for those peptide-HLA (pHLA) combinations[35].

We verified the HLA*07:02 restriction of peptides NP$_{233-244}$ (response in 9/10 donors), NP$_{30-38}$ (7/10 donors), NS1$_{196-206}$ (6/10 donors) and HA$_{255-263}$ (2/10 donors), but detected no responses towards NS2$_{62-70}$ (Fig. 4b). We verified the HLA-B*08:01 restriction for IBV-derived NP$_{92-99}$ (6/9 donors) and NP$_{479-487}$ (4/9 donors) peptides and confirmed the HLA-B*08:01 restriction for NP$_{30-38}$ (5/9 donors) (Fig. 4c). No responses were identified against NP$_{468-476}$. HLA-B*35:01-restriction was verified for NS1$_{260-268}$ (5/6 donors), HA$_{231-239}$ (3/6 donors) and PB2$_{66-70}$ (1/6 donors) peptides, while no responses were detected for NP$_{468-476}$ peptide (Fig. 4d). Weak IFN-γ$^+$TNF$^+$CD8$^+$ T cell responses for NP$_{165-173}$ were only observed in HLA-B*35:01 donors who co-expressed HLA-A*24:02 (A5 and A17; Supplementary Table 2). We hypothesize that HLA-A*24:02 co-expression may result in HLA-B*35:01/NP$_{165-173}$ (B35/NP$_{165-173}$)-cross-reactive CD8$^+$ T cell responses. Therefore, NP$_{165-173}$ was not considered as a true HLA-B*35:01-restricted IBV epitope.

HLA-restriction was further validated using peptide pHLA-I tetramer staining for 7 IBV epitopes that elicited the strongest IFN-γ$^+$TNF$^+$ CD8$^+$ T cell responses (Fig. 4a–d) and displayed >99% conservation among IBVs circulating in the human population between 1940 and 2023 (Table 1 and Supplementary Data 2). This included 2 HLA-B*07:02 (B7/NP$_{30}$, B7/NS1$_{196}$), 3 HLA-B*08:01 (B8/NP$_{30}$, B8/NP$_{92}$, B8/NP$_{479}$) and 2 HLA-B*35:01 (B35/HA$_{231}$, B35/NS1$_{260}$) tetramers. Prominent tetramer-

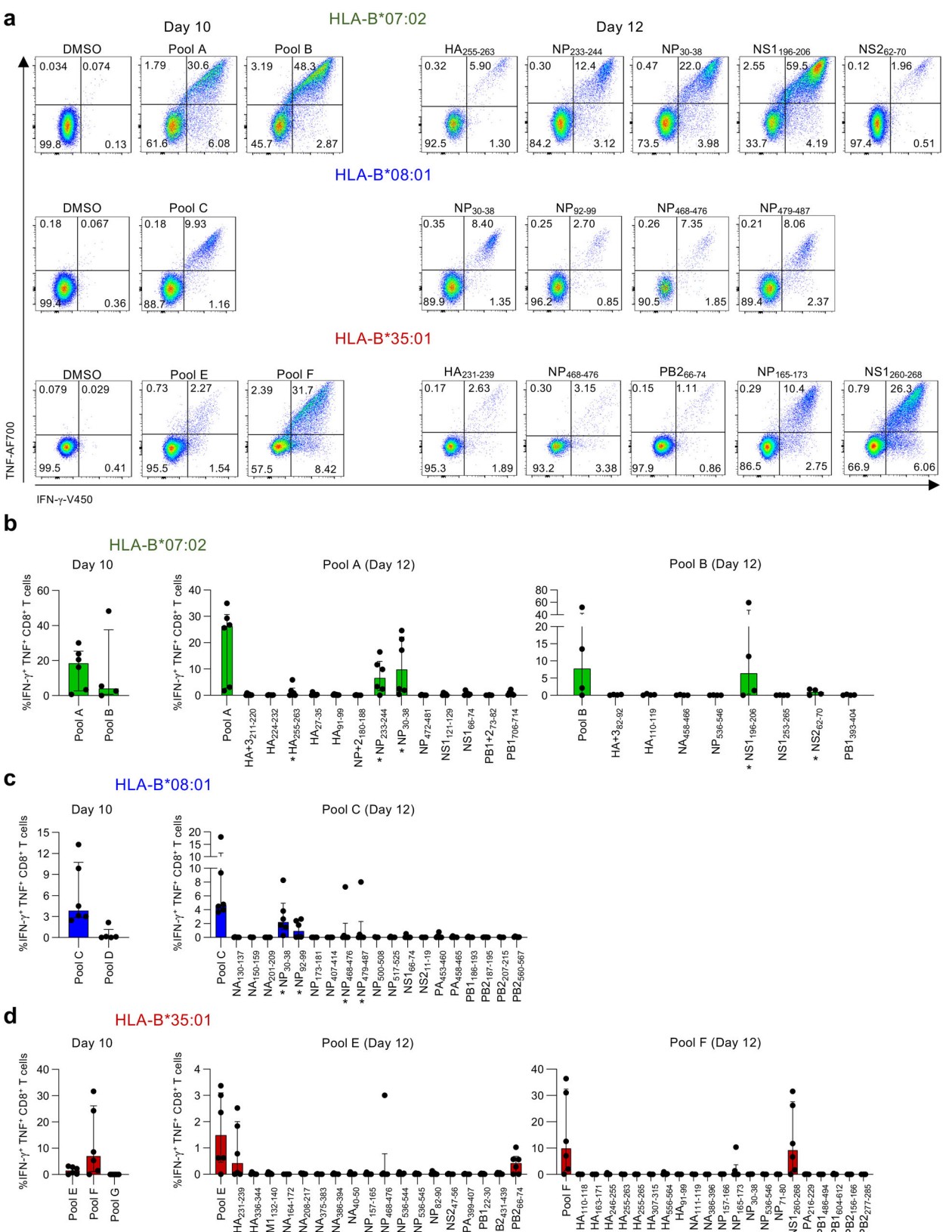

**Fig. 3 | CD8+ T cell reactivity to HLA-B*07:02-, HLA-B*08:01- and HLA-B*35:01-restricted IBV peptides. a** Representative FACS plots of IFN-γ+TNF+ CD8+ T cells restimulated with peptide pools or individual IBV peptides. **b** Frequency of IFN-γ+TNF+ CD8+ T cells after restimulation with HLA-B*07:02-restricted peptide pool A (*n* = 6 donors) or B (*n* = 4 donors) and individual HLA-B*07:02 IBV peptides (*n* = 4 donors for pool A and *n* = 6 donors for pool B). **c** Frequency of IFN-γ+TNF+ CD8+

T cells after restimulation with HLA-B*08:01-restricted pool C (*n* = 6 donors) or D (*n* = 5 donors) and individual HLA-B*08:01 IBV peptides (*n* = 6 donors). **d** Frequency of IFN-γ+TNF+ CD8+ T cells after restimulation with HLA-B*35:01-restricted pool E, F, G and individual HLA-B*35:01 IBV peptides (*n* = 6 donors). Median and interquartile range (IQR) are shown. DMSO background was subtracted. Source data are provided as a Source Data file.

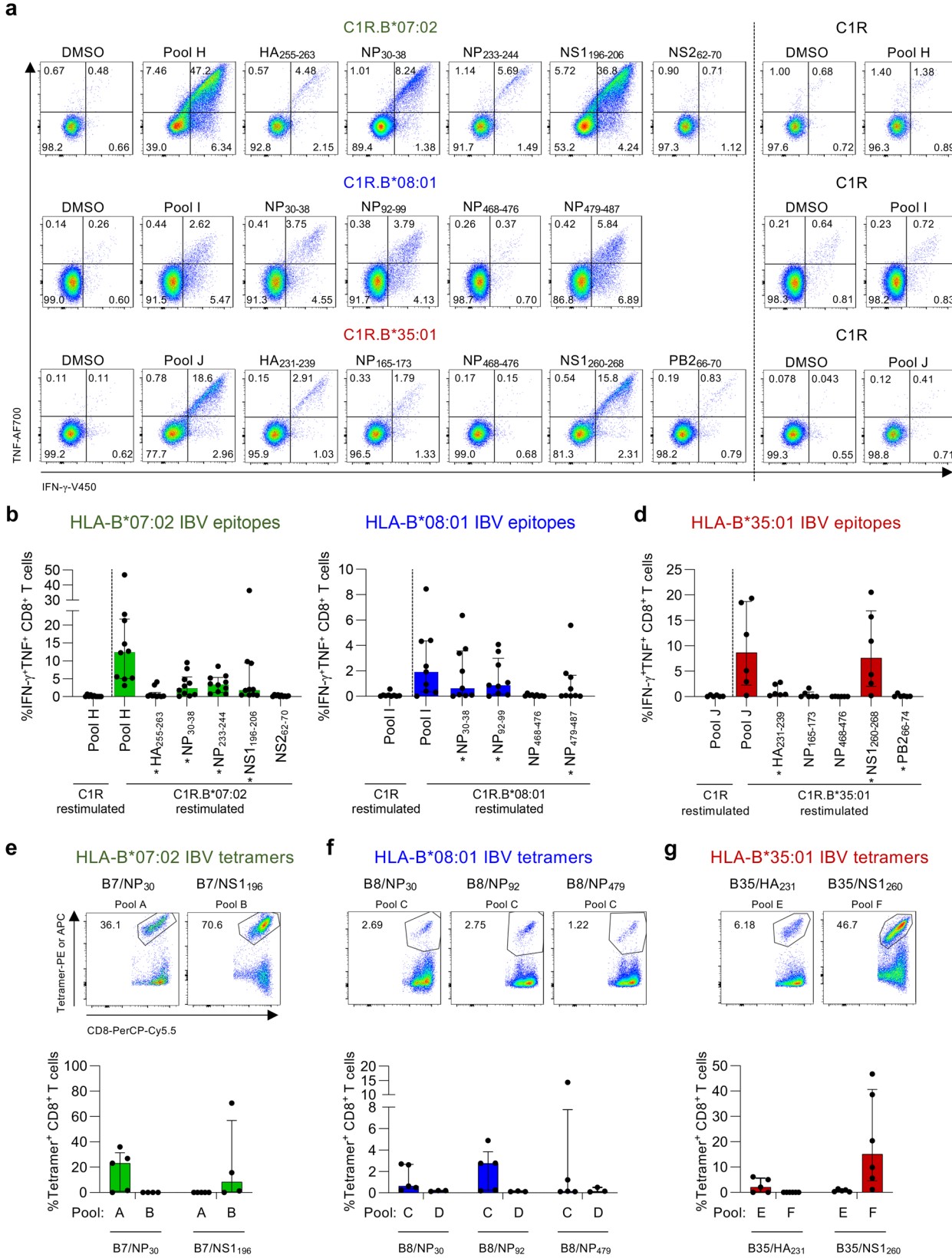

staining was observed in IBV-specific CD8$^+$ T cell lines that were expanded with their respective original peptide pool (A-G) (Fig. 4e–g). CD8$^+$ T cells expanded with an unrelated peptide pool were used as a negative control. B35/NS1$_{260}$-specific CD8$^+$ T cells had a higher frequency of tetramer$^+$CD8$^+$ T cells compared to the frequency of IFN-γ$^+$TNF$^+$ CD8$^+$ T cells following IBV peptide stimulation

(Supplementary Fig. 2b), suggesting that not all IBV epitope-specific CD8$^+$ T cells secrete both IFN-γ and TNF cytokines. Other IBV-specific CD8$^+$ T cell populations stained with their respective tetramers also displayed a similar trend.

Overall, we identified 9 IBV CD8$^+$ T cell epitopes, with 4 restricted to HLA-B*07:02, 2 restricted to HLA-B*08:01 and 3 restricted to

**Fig. 4 | CD8⁺ T cell responses towards HLA-B*07:02-, HLA-B*08:01 and HLA-B*35:01-restricted IBV epitopes. a** Representative FACS plots of IFN-γ and TNF production by CD8⁺ T cells after restimulation with C1R.B*07:02, C1R.B*08:01, C1R.B*35:01 or C1R cells pulsed with immunogenic IBV peptides or pools. Frequencies of IFN-γ⁺TNF⁺CD8⁺ T cells after restimulation with IBV peptide pulsed (**b**) C1R.B*07:02 (*n* = 10 donors), (**c**) C1R.B*08:01 (*n* = 9 donors) or (**d**) C1R.B*35:01 (*n* = 6 donors). **b**–**d** * indicate identified epitopes. DMSO background was subtracted. Frequencies and representative FACS plots of in vitro expanded IBV-specific CD8⁺

T cells stained with our generated (**e**) B7/NP₃₀₋₃₈ (named B7/NP₃₀) and B7/NS1₁₉₆₋₂₀₆ (B7/NS1₁₉₆) tetramers (*n* = 5 donors for pool A and *n* = 4 donors for pool B), (**f**) B8/NP₃₀₋₃₈ (B8/NP₃₀), B8/NP₉₂₋₉₉ (B8/NP₉₂) and B8/NP₄₇₉₋₅₈₇ (B8/NP₄₇₉) tetramers (*n* = 5 donors for pool C and *n* = 3 donors for pool D), and (**g**) B35/HA₂₃₁₋₂₃₉ (B35/HA₂₃₁) and B35/NS1₂₆₀₋₂₆₈ (B35/NS1₂₆₀) tetramers (*n* = 5 donors for pool E and *n* = 6 donors for pool F). Bars represent median and IQR. Overlapping peptides and peptide variants were counted as a single epitope (see "Methods"). Source data are provided as a Source Data file.

HLA-B*35:01, and confirmed the HLA-B*08:01 restriction of the previously identified NP₃₀₋₃₈. Our study increases the overall number of known IBV CD8⁺ T cell epitopes by 50% as well as covering 2 additional prominent HLAs, HLA-B*07:02 and HLA-B*35:01. Of note, our newly-identified IBV epitopes were not conserved among IAV strains as we were unable to identify overlapping sequences within IAV proteins.

### Robust ex vivo frequencies of IBV-specific CD8⁺ T memory populations in healthy donors

To probe the magnitude of IBV-specific CD8⁺ T cell populations directly ex vivo, we used tetramer-associated magnetic enrichment (TAME)[36] in PBMCs from 22 HLA-B*07:02⁺, 19 HLA-B*08:01⁺ and 11 HLA-B*35:01⁺ healthy donors (Fig. 5, Supplementary Fig. 3a and Supplementary Table 2). Substantial tetramer⁺CD8⁺ T cell populations were directed against B7/NP₃₀₋₃₈ (median frequency of $1.90 \times 10^{-5}$), B7/NS1₁₉₆₋₂₀₆ ($2.65 \times 10^{-5}$), B8/NP₃₀₋₃₈ ($9.50 \times 10^{-6}$), B8/NP₉₂₋₉₉ ($2.35 \times 10^{-5}$), B35/HA₂₃₁₋₂₃₉ ($1.70 \times 10^{-5}$) and B35/NS1₂₆₀₋₂₆₈ ($6.77 \times 10^{-5}$) epitopes (Fig. 5a, b). These magnitudes were comparable to previously published ex vivo IBV-specific CD8⁺ T cell frequencies for HLA-A*02:01- and HLA-A*24:02-restricted IBV-epitopes[24,26]. B35/NS1₂₆₀⁺CD8⁺ T cell frequencies ($6.77 \times 10^{-5}$) were higher than B8/NP₃₀⁺CD8⁺ ($9.50 \times 10^{-6}$) and B8/NP₄₇₉⁺CD8⁺ ($2.74 \times 10^{-6}$) T cell frequencies ($p = 0.0145$ and $p = 0.0017$, respectively). Low prevalence of B8/NP₄₇₉⁺CD8⁺ T cells was evident and 7/11 donors had <10 B8/NP₄₇₉⁺CD8⁺ T cells, thus this epitope was not studied beyond phenotypic analysis.

To define ex vivo phenotype of IBV-specific tetramer⁺CD8⁺ T cells, we analyzed expression of CD27, CD45RA and CD95 to discriminate between central memory (T_cm; CD27⁺CD45RA⁻), effector memory (T_em; CD27⁻CD45RA⁻), terminally differentiated effector memory (T_emra; CD27⁻CD45RA⁺), stem cell memory (T_scm; CD27⁺CD45RA⁺CD95⁺) and naive (T_naive; CD27⁺CD45RA⁺CD95⁻) CD8⁺ T cell phenotypes (Fig. 5c). Cell populations with <10 acquired tetramer⁺CD8⁺ T cells were excluded from phenotypic analyses[37]. T_cm phenotype was more enriched in

IBV-specific CD8⁺ T cells compared to total CD8⁺ T cells, whereas T_emra phenotype was less prominent (Fig. 5d).

Overall, our findings provide evidence for substantial ex vivo pools of IBV-specific tetramer⁺CD8⁺ T cells circulating in peripheral blood and expressing predominantly memory T_cm phenotype in healthy adults.

### IBV-specific CD8⁺ T cell responses decline with age

CD8⁺ T cells directed towards IAV-derived epitopes substantially change throughout the human lifespan at numerical, phenotypic and transcriptomic levels[38–40]. An unanswered question remains how IBV-specific CD8⁺ T cells change throughout immunologically-distinct phases of life, especially given age-related IBV disease severity[7–9]. We sought to define the magnitude and phenotype of IBV-specific CD8⁺ T cells across three age groups (6 children <18 years; 28 adults 18–59 years; 9 elderly adults >60 years) (Fig. 6a). Pooled analysis revealed that IBV-specific CD8⁺ T cell responses peaked in adults and declined in the elderly (Fig. 6b). Similar trends were observed within HLA-B*07:02⁺ donors and HLA-B*08:01⁺ donors but could not be confirmed in HLA-B*35:01⁺ individuals, due to the lack of available elderly HLA-B*35:01⁺ donors (Fig. 6c).

Phenotypic analysis revealed that T_cm frequencies of pooled IBV-specific tetramer⁺CD8⁺ T cells increased with age and peaked in the elderly (median T_cm frequency in children 27.4%, adults 61.9%, elderly 75.2%) (Fig. 6d), which was also observed for HLA-B*07:02-restricted IBV tetramer⁺CD8⁺ T cells (Supplementary Fig. 3b). This was in agreement with our recent observation for IAV-specific CD8⁺ T cells directed at the HLA-A*02:01-restricted M1₅₈₋₆₆ (A2/M1₅₈) epitope[41]. In contrast, total CD8⁺ T_cm populations were stable with age while the T_naive population decreased with age and T_emra/T_em cells increased (Supplementary Fig. 3c), in line with previous reports[39,41].

In summary, despite maintaining a predominant T_cm phenotype across the human lifespan, IBV-specific CD8⁺ T cell populations decline in magnitude with age.

### IBV-specific CD8⁺ T cells in human tissues express a tissue resident phenotype

Thus far, we have focussed on IBV-specific CD8⁺ T cells within peripheral blood. However, tissue-resident memory T (T_rm) cell populations are important for protective immunity, especially towards respiratory viral infections[42]. Mouse models show lung-resident T cells protect against severe influenza disease outcomes[43,44]. We thus assessed the prevalence and phenotype of IBV-specific CD8⁺ T cells in human lungs (*n* = 6 donors), mesenteric lymph nodes (LN; *n* = 11) and spleens (*n* = 15) from deceased donors and tonsils (*n* = 15) from routine tonsillectomies (Supplementary Table 2).

Median frequencies of pooled IBV tetramer⁺CD8⁺ T cells ranged between $1.63 \times 10^{-5}$ and $1.40 \times 10^{-4}$ in human tissues, with higher frequencies in LN ($1.40 \times 10^{-4}$) and tonsils ($3.84 \times 10^{-5}$) compared to spleen ($1.63 \times 10^{-5}$) (Fig. 7a). LN frequencies were also higher than PBMCs ($2.56 \times 10^{-5}$). Similar trends were observed for HLA-B*07:02-restricted IBV tetramer⁺CD8⁺ T cells, albeit not significant.

To establish whether IBV tetramer⁺CD8⁺ T cells in human tissues displayed a T_rm phenotype, we assessed CD69 and CD103 expression on IBV tetramer⁺CD8⁺ T cells in lungs, secondary lymphoid organs

### Table 1 | Conservation of IBV-derived immunogenic peptides

| Peptide | Year period | HLA restriction | Sequence | Overall conservation (%) |
|---|---|---|---|---|
| HA₂₅₅₋₂₆₃ | 1940–2022 | HLA-B*07:02 | LPQSGRIVV | 99.9 |
| NP₃₀₋₃₈ | 1940–2020 | HLA-B*07:02 and HLA-B*08:01 | RPIIRPATL | 99.94 |
| NS1₁₉₆₋₂₀₆ | 1940–2017 | HLA-B*07:02 | HPNGYKSLSTL | 99.58 |
| NP₂₃₃₋₂₄₄ | 1940–2020 | HLA-B*07:02 | LPRRSGATGVAI | 91.91 |
| NP₉₂₋₉₉ | 1940–2020 | HLA-B*08:01 | QMMVKAGL | 99.98 |
| NP₄₇₉₋₄₈₇ | 1940–2020 | HLA-B*08:01 | QAVRRMLSM | 99.9 |
| HA₂₃₁₋₂₃₉ | 1940–2022 | HLA-B*35:01 | SANGVTTHY | 99.7 |
| NS1₂₆₀₋₂₆₈ | 1940–2017 | HLA-B*35:01 | TAVGVLSQF | 99.49 |
| PB2₆₆₋₇₀ | 1940–2022 | HLA-B*35:01 | MANRIPLEY | 99.84 |

All amino acid sequences present in the National Center for Biotechnology Information (NCBI; http://www.ncbi.nlm.nih.gov/genomes/FLU) database at 9 September 2022 (HA), 26 November 2021 (NP) or 23 March 2023 (NS1).

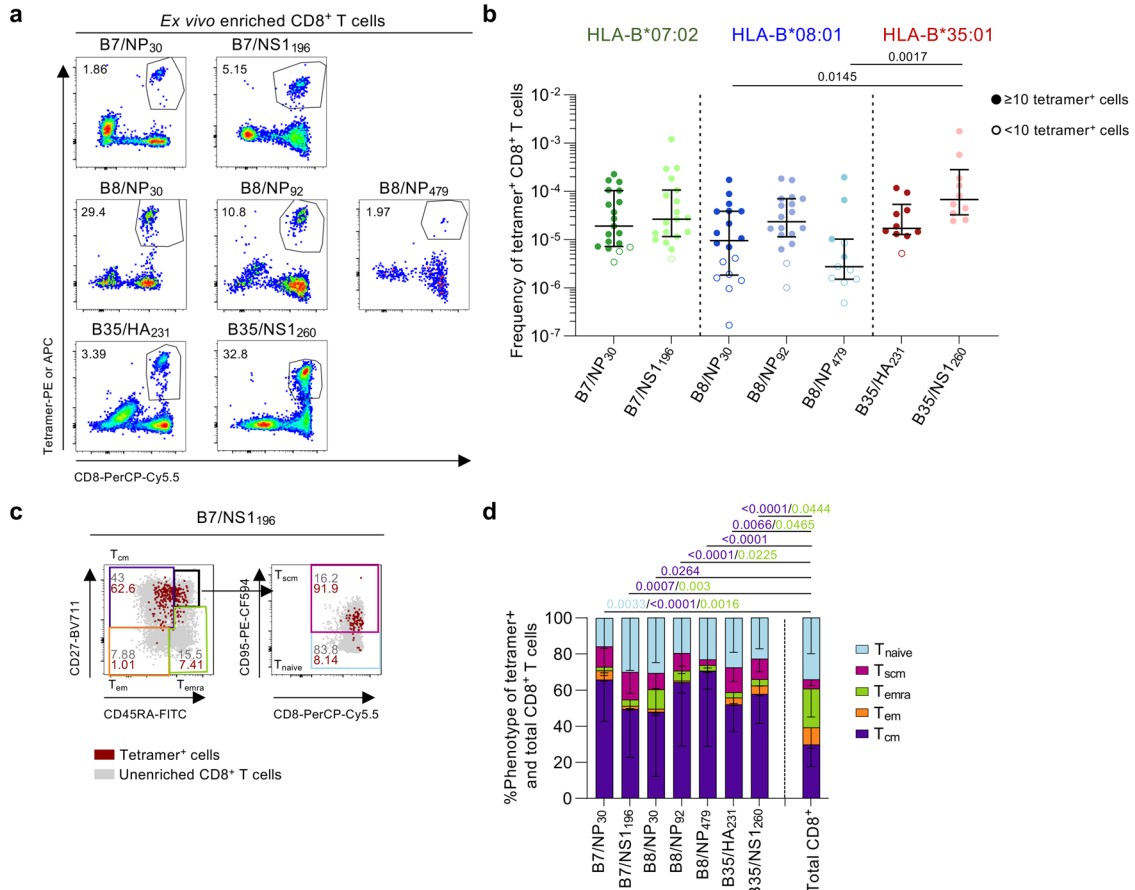

**Fig. 5 | Ex vivo detection of IBV-specific CD8⁺ T cells in healthy PBMCs.**
**a** Representative FACS plots of enriched IBV-specific CD8⁺ T cells. **b** Frequency of B7/NP₃₀⁺CD8⁺ T cells (*n* = 19 donors), B7/NS1₁₉₆⁺CD8⁺ T cells (*n* = 19), B8/NP₃₀⁺CD8⁺ T cells (*n* = 18), B8/NP₉₂⁺CD8⁺ T cells (*n* = 18), B8/NP₄₇₉⁺CD8⁺ T cells (*n* = 11), B35/HA₂₃₁⁺CD8⁺ T cells (*n* = 10 donors) and B35/NS1₂₆₀⁺CD8⁺ T cells (*n* = 10). Open symbols represent <10 tetramer⁺ cells counted, excluded from phenotypic analysis. Bars represent median and IQR. **c** Representative FACS plots of the gating strategy to identify phenotypic populations T_cm (CD27⁺CD45RA⁻) cells, T_em (CD27⁻CD45RA⁻), T_emra (CD27⁻CD45RA⁺), T_naïve (CD27⁺CD45RA⁺CD95⁻) and T_scm

(CD27⁺CD45RA⁺CD95⁺) cells. Gray dots represent total CD8⁺ T cells in the unen-riched sample, red dots are IBV-specific CD8⁺ T cells in the enriched sample.
**d** Proportion of memory phenotypes for each IBV-specific CD8⁺ T cell population and total CD8⁺ T cells (*n* = 16 donors; B7/NP₃₀, *n* = 18; B7/NS1₁₉₆, *n* = 11; B8/NP₃₀, *n* = 16; B8/NP₉₂, *n* = 4; B8/NP₄₇₉, *n* = 9; B35/HA₂₃₁, *n* = 10; B35/NS1₂₆₀, *n* = 43; Total CD8⁺ T cells). Bars represent mean and standard deviation (SD). Statistical sig-nificance was determined with using a two-sided Kruskal–Wallis with Dunn's test for multiple comparisons (**b**) or two-way ANOVA with two-sided Tukey's test for mul-tiple comparisons (**d**). Source data are provided as a Source Data file.

(SLOs; tonsils and LN) and spleens (Fig. 7b). Higher proportions of CD69⁺CD103⁺ and CD69⁺CD103⁻ T_rm were detected within the pooled IBV tetramer⁺CD8⁺ T cell population in lungs (29.9% and 34.4%, respectively) compared to spleen (12.3% and 18.1%, respectively), which was associated with lower CD69⁻CD103⁻ subset in lungs (33.1%) compared to spleens (65.8%) and secondary lymphoid organs (SLOs; 56.4%) (Fig. 7c). Similar trends were observed for individual IBV tetramer⁺CD8⁺ T cells albeit not significant, except for B8/NP₉₂⁺CD8⁺ T cells.

Higher T_em populations in pooled and individual IBV tetramer⁺CD8⁺ T cells were mainly observed in the lung and spleen compared to circulating IBV tetramer⁺CD8⁺ T cells, while T_naïve popu-lations were mainly detected in SLOs and in circulation (Fig. 7d), in line with the expected trafficking patterns of these memory T cell subsets[45,46]. When analyzing individual tetramer⁺CD8⁺ T cells, the pat-tern of memory phenotypes for each tissue matched the overall IBV-specific CD8⁺ T cell population.

LN and spleen samples obtained from the same individuals (*n* = 12 donors) were analyzed to assess organ-related differences in IBV-specific CD8⁺ T cell frequency and phenotype within these individuals (Supplementary Fig. 4). Higher frequencies of IBV tetramer⁺CD8⁺ T cells, both pooled and individual responses, were detected in LN

(median 1.4 × 10⁻⁴) compared to spleens (median 1.52 × 10⁻⁵). Likewise, proportions of IBV-specific CD69⁺CD103⁺CD8⁺ T_rm were higher in the LN (median 19.7%) compared to spleens (median 5.0%), albeit not significant.

Overall, we identified prominent IBV tetramer⁺CD8⁺ T_rm cell populations in five human tissues demonstrating that IBV tetramer⁺CD8⁺ T cells are poised to provide protection at the site of IBV infection (lung). We also show that IBV-specific tetramer⁺CD8⁺ T cells are found at a higher frequency in LNs compared to spleen but are not necessarily resident within the LN.

## B*07:02-restricted 11 amino acid long NS1₁₉₆₋₂₀₆ peptide bulges out of the peptide-binding cleft

Although representing approximately 11% of HLA-B*07:02 ligands (Fig. 2a), 11mer peptides such as, the NS1₁₉₆₋₂₀₆ peptide are considered long with regards to the structural constraints of the HLA-I binding cleft[47]. Long peptides can either have N or C terminal overhangs[48] or bulge out of the HLA-peptide binding cleft eliciting either highly restricted[49] or diverse TCR repertoires[50], which can be associated with lower epitope-specific CD8⁺ T cell frequencies when associated with higher mobility[32]. To establish whether the NS1₁₉₆₋₂₀₆ peptide also bulges out of the HLA-B*07:02 peptide binding cleft, we crystallized

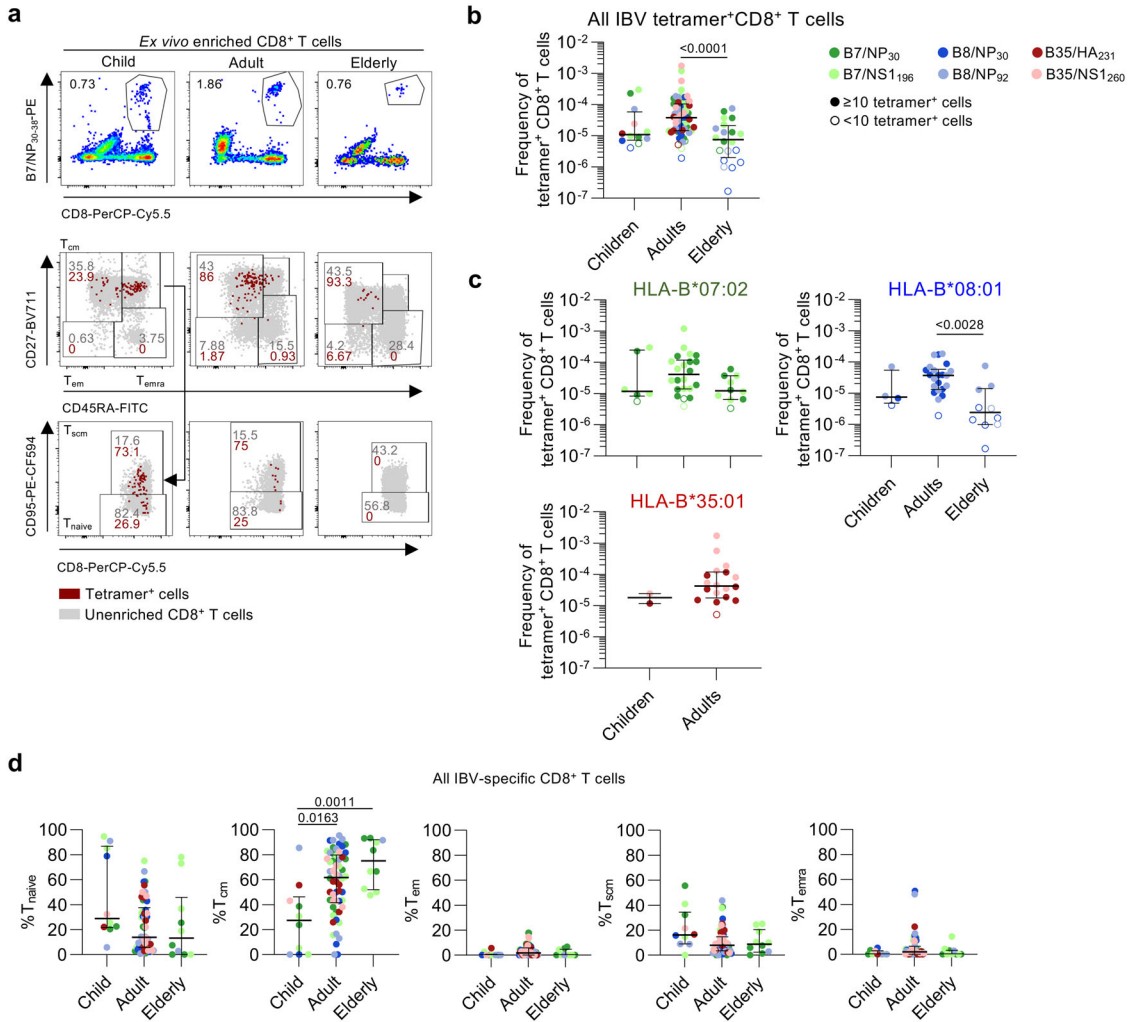

**Fig. 6 | Age-associated changes to IBV-specific CD8$^+$ T cells. a** Representative FACS plots of B7/NP$_{30}$CD8$^+$ T cells across age. **b** Frequency of pooled IBV-specific CD8$^+$ T cells across age groups ($n = 43$ donors). Colors represent different tetramers. Open symbols represent <10 tetramer$^+$ cells counted. **c** Frequency of HLA-B$^*$07:02 ($n = 19$ donors), -B08:01 ($n = 18$) and -B35:01-restricted ($n = 10$) IBV-specific CD8$^+$ T cells across age grouped by HLA ($n = 10–19$ donors). Colors represent different tetramers. Open symbols represent <10 tetramer$^+$ cells counted.

**d** Proportions of phenotypes of IBV-specific CD8$^+$ T cells in children, adults and elderly donors for all IBV tetramer$^+$ populations ($n = 38$ donors). **b–d** Statistical significance was determined using a two-sided Kruskal–Wallis with Dunn's test for multiple comparisons for pooled and HLA-B$^*$07:02 and HLA-B$^*$08:01 tetramers and (**c**) two-sided Mann–Whitney $U$ test for HLA-B$^*$35:01 tetramers. Bars represent median and IQR. Source data are provided as a Source Data file.

and solved the structure of HLA-B$^*$07:02-NS1$_{196-206}$ at a resolution of 2.4 Å (Fig. 8a and Supplementary Table 4). Structural analysis showed that the peptide NS1$_{196-206}$ interacts with hydrogen bonds to side chain residues of HLA α1 and α2 helices. Considering the anchor residues at the extremities of the peptide (P2 and PΩ), the 11-mer peptide bulged out from central P4-Gly to P8-Leu to adopt stable conformation and providing a surface for TCR recognition. In terms of HLA-I and peptide specificity, polymorphic residues Asp-114, and Glu-152 form hydrogen bonds with the side chain of P6-Lys and P9-Ser, respectively (Fig. 8a). Accordingly, the NS1$_{196-206}$ epitope forms a stable conformation that bulges out of the HLA-B$^*$07:02 cleft, which is in accordance with the relatively robust B7/NS1$_{196}$-specific CD8$^+$ T cell frequencies (Fig. 5b).

**Presentation of NP$_{30-38}$ in the context of HLA-B$^*$07:02**
We sought to assess whether structural differences of NP$_{30-38}$ bound by HLA-B$^*$07:02 or HLA-B$^*$08:01 allomorphs could underpin similarities or differences in the responding TCR repertoires. Antigen presentation of NP$_{30-38}$ (RPIIRPATL) peptide bound to HLA-B$^*$07:02-NP$_{30-38}$ was assessed after obtaining a 2.2 Å resolution structure (Fig. 8b, c). Canonical TCR-docking modes often focus heavily near the mid-point

of the α1/2 helices such that central epitope residues play a key role in TCR recognition. Within the HLA-B$^*$07:02-NP$_{30-38}$ complex the P5-Arg is directed towards the HLA platform leaving the P3 and P4-Ile and P6-Pro as likely determinants for T cell engagement (Fig. 8b). HLA-B$^*$07:02 shares 95% sequence identity with HLA-B$^*$08:01, but a significant point-of-difference is an Asp114Asn polymorphism. Superposing the HLA-B$^*$08:01 structure with HLA-B$^*$07:02-NP$_{30-38}$ indicates that Asn-114 is capable of accepting the platform-directed P5-Arg within HLA B$^*$08:01 (Fig. 8c) with minimal changes.

Since NP$_{30-38}$ has anchor residues that match the HLA-B$^*$35:01 binding motifs (Fig. 2b), we sought a structural explanation of why we could not detect HLA-B35:01/NP$_{30}$CD8$^+$ T cell responses (Fig. 3d). Superposing HLA-B$^*$35:01 with HLA-B$^*$07:02-NP$_{30-38}$ shows that presence of the Arg97 polymorphism within the HLA-B$^*$35:01 allomorph would align a positively charged HLA residue at a similar position to that of P5-Arg from within the NP$_{30-38}$ epitope. Repulsion of like-charges may limit peptide binding (Fig. 8c) or require significant changes to epitope conformation for this pHLA combination.

Overall, the structural analyses indicated the NP$_{30-38}$ peptide can be accommodated with both HLA-B$^*$07:02 and HLA-B$^*$08:01.

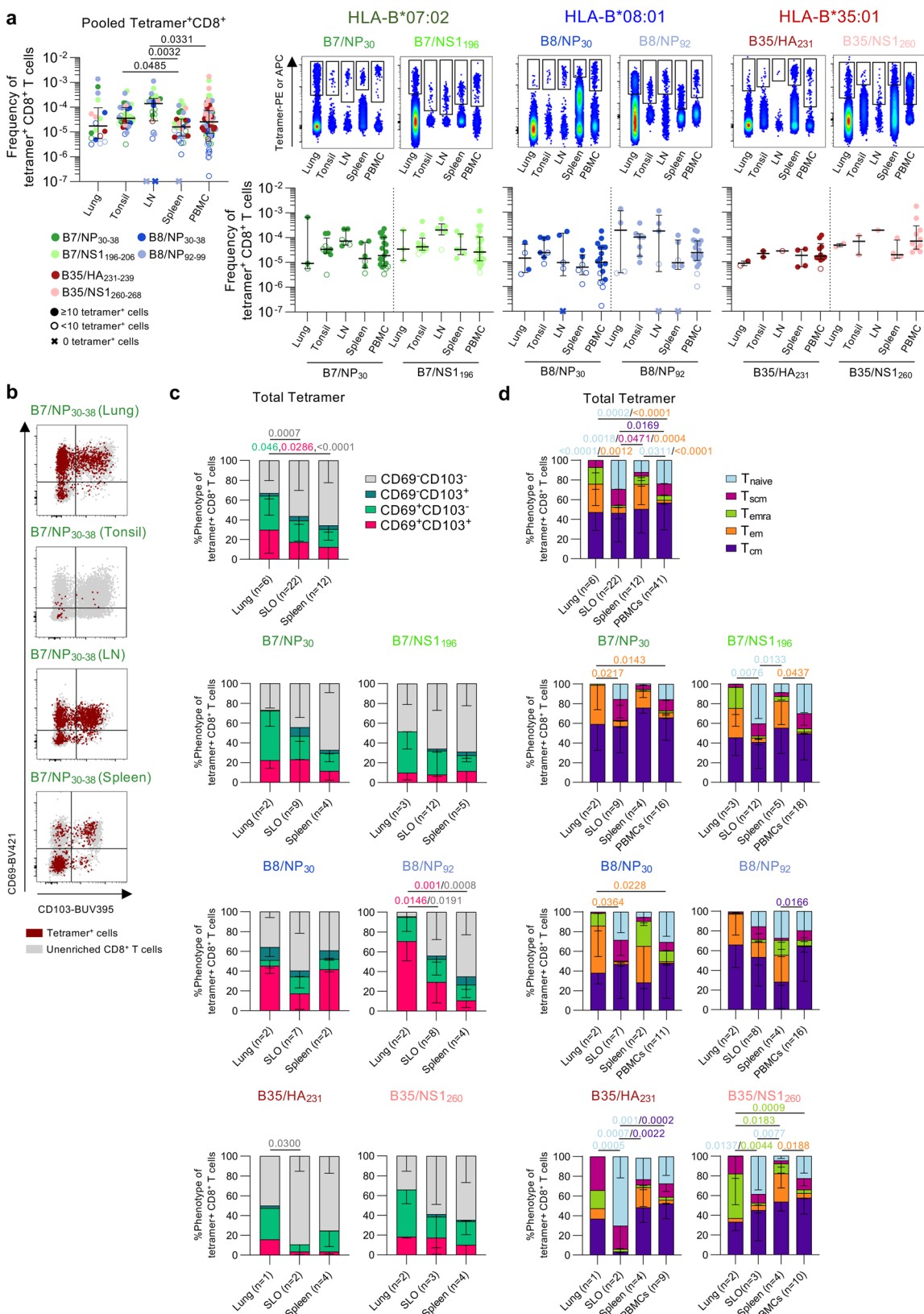

However, allele specific polymorphisms may result in structural differences of $NP_{30-38}$ presentation depending on the allomorph. Our data also suggested that $NP_{30-38}$ would form a stable interaction with HLA-B*35:01, providing a potential explanation as to why no HLA-B*35:01/$NP_{30}^+$ CD8$^+$ T cell response was detected against this allomorph.

## Distinct TCRαβ repertoires are elicited towards B7/$NP_{30}$ and B8/$NP_{30}$ epitopes

Given the presentation of the IBV $NP_{30-38}$ peptide by HLA-B7:02 and HLA-B*08:01, we asked whether B7/$NP_{30}^+$CD8$^+$ and B8/$NP_{30}^+$ T cell responses consisted of cross-reactive TCRαβ clonotypes. To this end, we co-stained PBMCs obtained from HLA-B*07:02 and

**Fig. 7 | IBV-specific CD8⁺ T cells across human tissues. a** Frequencies of total IBV-specific CD8⁺ T cells in human lungs ($n = 6$ donors), tonsils ($n = 15$), lymph nodes ($n = 11$), spleen ($n = 15$) and PBMCs ($n = 43$). Representative concatenated FACS plots of enriched IBV-specific CD8⁺ T cells and frequency of individual tetramer⁺CD8⁺ T cells across PBMCs, LN, tonsils, spleen and lung. Open symbols represent <10 tetramer⁺ cells counted. "x" symbols represent 0 tetramer⁺ cells. **b** Representative FACS plots of CD69 and CD103 expression in IBV-specific CD8⁺ T cells from different tissues. **c** Proportions of CD69 and CD103 expressing subsets for total IBV-specific CD8⁺ T cells ($n = 36$ donors) and individual tetramer⁺CD8⁺ T cell populations across different tissues (lungs, SLOs and spleen). **d** Proportions of memory phenotypes of total IBV-specific CD8⁺ T cells ($n = 36$ donors) and individual tetramer⁺CD8⁺ T cell populations across different tissues. **a** Bars represent median and IQR. **c**, **d** Bars represent mean and SD. Statistical significance was determined using a two-sided Kruskal–Wallis with Dunn's test for multiple comparisons (**a**) or two-way ANOVA with two-sided Tukey's test for multiple comparisons (**c**, **d**). Source data are provided as a Source Data file.

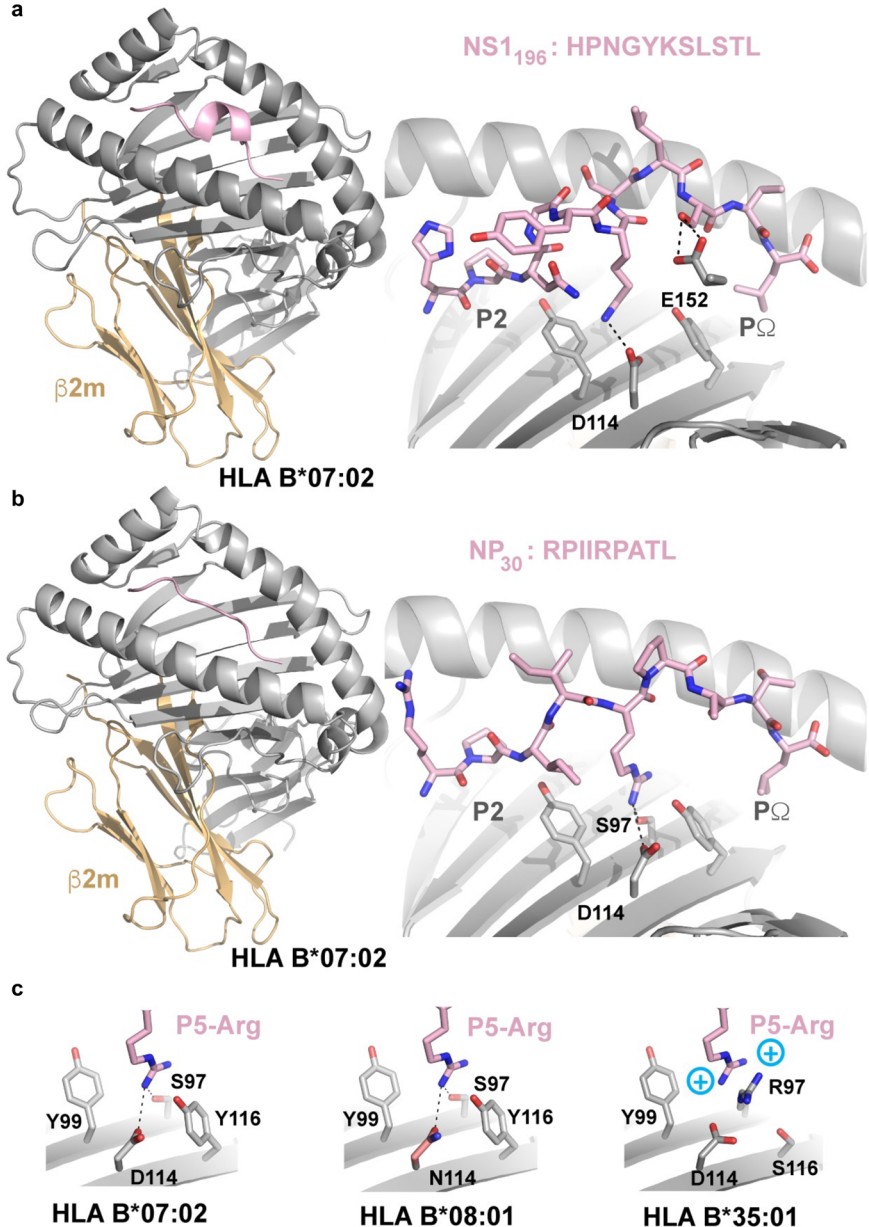

**Fig. 8 | Crystal structures of IBV peptides presented by HLA-B*07:02. a** Crystal structure of HLA-B*07:02-NS1196-206 where HLA-B*07:02 is represented as an gray/orange cartoon while NS1196-206 is represented as pink sticks. **b** Cartoon representation of the crystal structure of the NP30-38 peptide when presented by HLA-B*07:02 (color coding as in (**a**)) with the positions of the P2 and PΩ labeled. Key residues interacting with the P5-Arg are labeled. **c** Overlay of the P5-Arg pockets in HLA-B*07:02, -B*08:01 and -B*35:01 demonstrating polymorphisms incompatible with the downward P5-Arg.

HLA-B*08:01 double-positive and single-positive individuals with B7/NP30-PE and B8/NP30-APC tetramers (Fig. 9a). Although we identified single-positive B7/NP30⁺CD8⁺ and/or B8/NP30⁺CD8⁺ T cells across all individuals, we did not detect any cross-reactive double-positive B7/NP30⁺B8/NP30⁺CD8⁺ T cells. This suggests a lack of shared TCRs directed towards B7/NP30⁺CD8⁺ and B8/NP30⁺CD8⁺ T cells. We identified no significant differences in B7/NP30⁺ or B8/NP30⁺ CD8⁺ T cell frequencies between HLA-B*07:02 and HLA-B*08:01 single-positive

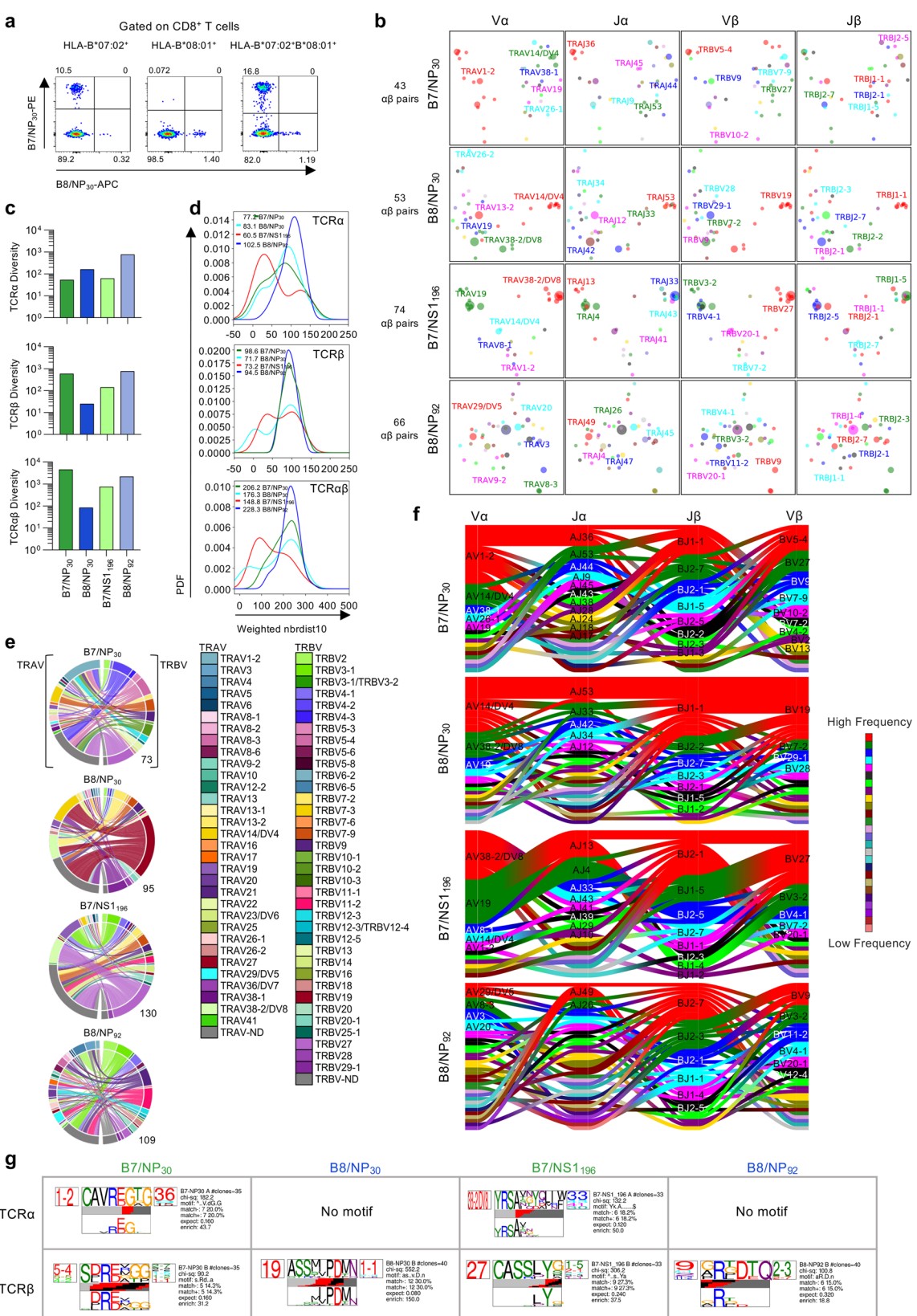

and double-positive individuals, however a trend for lower frequencies was observed in HLA-B*08:01-expressing donors (Supplementary Fig. 5a). A phenotypic comparison of B7/NP$_{30}$- and B8/NP$_{30}$-specific CD8$^+$ T cell populations in HLA-B*07:02 and HLA-B*08:01 single-positive and double-positive individuals revealed a prominent B7/NP$_{30}$ T$_{cm}$ population in HLA-B*07:02-expressing donors regardless of

HLA-B*08:01 co-expression, while a visually apparent but statistically indistinguishable trend for higher B8/NP$_{30}$ T$_{cm}$ frequencies was observed for HLA-B*08:01-expressing donors versus double-positive donors (Supplementary Fig. 5b). These results indicate that co-expression of HLA-B*07:02 and HLA-B*08:01 may result in lower recruitment of B8/NP$_{30}^+$CD8$^+$ T cells, which could potentially be

**Fig. 9 | TCRαβ repertoire of IBV-specific CD8⁺ T cells. a** FACS plots of HLA-B*07:02⁺HLA-B*08:01⁺ individuals after dual B7/NP₃₀ and B8/NP₃₀ tetramer staining and enrichment. **b** 2D kernel principal component analysis (kPCA) projections of B7/NP₃₀⁺CD8⁺, B8/NP₃₀⁺CD8⁺, B7/NS1₁₉₆⁺CD8⁺ and B8/NP₉₂⁺CD8⁺ TCR landscapes colored by Vα, Jα, Vβ and Jβ gene usage (left to right). **c** TCRdiv diversity measures of the TCRα, TCRβ or paired TCRαβ chains. **d** Smoothed density profiles of neighbor distance distribution are shown for each epitope. Lower peaks indicate more clustered IBV-specific CD8⁺ single TCRα, TCRβ or paired TCRαβ repertoire, with average distance values depicted within the plot. PFD stands for probability density function. **e** Circos plots of TRAV and TRBV gene usage per IBV-specific CD8⁺ T cell population. Left arch segment colors indicate TRAV usage, right outer arch colors depict TRBV usage. Connecting lines indicated TRAV-TRBV gene pairing and are colored based on their TRBV usage and segmented based on their CRD3α and CDR3β sequence, the thickness is proportional to the number of TCR clones with the respective pair. The number of sequences considered for each circos plot is shown at the right bottom. **f** Gene segment usage and gene-gene pairing landscapes, vertical stacks indicate V and J segments and gene pairing indicated by curved paths, thickness is proportional to the number of TCR clones within the respective gene pairing. Genes are colored by frequency within the repertoire. **g** Top-scoring IBV-specific CD8⁺ CDR3α (top TCR logo) and CDR3β (bottom TCR logo) sequence motifs for each IBV-specific CD8⁺ T cell population (columns). Each logo depicts the V- (left side) and J- (right side) gene frequencies with the CDR3 amino acid sequence in the middle with the full height (top) and scaled (bottom) by per-residue reparative entropy to background frequencies derived from TCRs with matching gene-segment composition to highlight motif positions under selection. The middle section indicates the inferred rearrangement structure by source region (light gray for V-region, dark gray for J, black for D and red for N-insertions) of the grouped receptors. Motif scores were determined by chi-squared, with values above 90 considered significant. Source data are provided as a Source Data file.

explained by the slightly reduced binding affinity of NP₃₀₋₃₈ to HLA-B*08:01 compared to HLA-B*07:02 (Supplementary Data 1). Since TCRαβ repertoire diversity and clonal composition affect functionality of epitope-specific CD8⁺ T cells[32,41,51], we dissected the B7/NP₃₀⁺CD8⁺ and B8/NP₃₀⁺CD8⁺ TCRαβ repertoires and compared those with TCRαβ signatures within other immunodominant HLA*B*07:02- and HLA-B*08:01-restricted CD8⁺ T cell epitopes, namely B7/NS1₁₉₆₋₂₀₆ and B8/NP₉₂₋₉₉. Overall, we analyzed 236 paired TCRαβ clonotypes, 23 single TCRα and 148 TCRβ chains. For each epitope, an average of 55 TCRαβ clonotypes were analyzed across 5-6 donors (Supplementary Tables 6–9). 2D kernel principal component analysis (kPCA) projections of the TCR Vα, Vβ, Jα and Jβ gene segment landscape revealed that the overall TCRαβ repertoire of HLA-B*07:02- and HLA-B*08:01-restricted IBV epitopes was highly diverse (Fig. 9b).

This was also reflected by high TCR diversity scores (TCRdiv) for single TCRα, TCRβ and paired TCRαβ repertoires relative to B8/NP₃₀ (Fig. 9c). Neighbor distance distributions were calculated to measure the density within each epitope-specific TCR repertoire and quantify the relative contribution of clustered and diverged TCRs. Lower average values of the distance distribution peak represented a more similar clustering of clonotypes (Fig. 9d). The level of clustering greatly varied between the studied IBV epitope-specific CD8⁺ T cell populations. A bimodal distribution that was mainly driven by the TCRβ chain was observed for B8/NP₃₀, whereas the bimodal distribution of B7/NS1₁₉₆ was driven by both the TCRα and β chain. In contrast, B7/NP₃₀ TCRαβ had an overall lower similarity in clustering of the paired TCRαβ clonotypes, which was mainly driven by high diversity in the TCRβ chain. Highest diversity was however observed for B8/NP₉₂ (Fig. 9d). Overall, 2–8 clonotypes were expanded per donor per epitope (Supplementary Tables 6–9). Very little clonotype sharing was observed, indicative of private TCR repertoires.

To understand whether the observed differences in TCRαβ diversity were related to gene segment usage, V and J genes (Vα, Vβ, Jα, Jβ) were analyzed for each epitope (Fig. 9e, f and Supplementary Fig. 5c). B7/NP₃₀⁺ TCRαβ repertoire revealed a preference for the TRAV1-2 gene segment and a diverse TRBV gene usage, consistent with our finding of the TCRα chain driving clustering in our neighbor distance analysis. TRAV1-2 usage was observed in 5/6 donors (regardless of whether they co-expressed HLA-B*07:02 and HLA-B*08:01) but was particularly prominent in two donors (A10 and A27). B8/NP₃₀⁺CD8⁺ TCRαβ repertoire was dominated by TRBV19-expressing clonotypes detected in all 5 donors and was frequently paired with TRBJ1-1 and TRAV14/DV4 and diverse TRAJ usage. The B8/NP₃₀⁺ CD8⁺ TCRαβ repertoire was more diverse in donors who co-expressed HLA-B*07:02, suggesting that co-expression of these alleles may alter the B8/NP₃₀⁺ TCRαβ repertoire[52,53]. B7/NS1₁₉₆⁺CD8⁺ TCRαβ repertoire was dominated by TRBV27-TRAV38-2/DV8 expressing clonotypes paired with a variety of TRAJ and TRBJ gene segments and was detected in 3/6 donors (Fig. 9e, f and Supplementary Fig. 5c). TRAV19-expressing

clonotypes were also observed in all donors but paired with different TRAJ, TRBV and TRBJ gene segments, which matched the two clusters in the kPCA projections. In line with the kPCA and neighbor distance distribution analysis, the B8/NP₉₂-specific TCRαβ repertoire was highly diverse in gene usage.

In summary, all IBV-specific TCRαβ repertoires were relatively diverse, with pockets of clustering clonotypes observed for B7/NP₃₀, B8/NP₃₀ and B7/NS1₁₉₆, but not B8/NP₉₂-specific T cells. Furthermore, B7/NP₃₀⁺CD8⁺ and B8/NP₃₀⁺CD8⁺ T cells have vastly distinct pHLA-I specificities as confirmed by their different gene segments and distinct TCRαβ repertoires.

## Distinct CDR3αβ motifs of HLA-B*07:02 and HLA-B*08:01-restricted CD8⁺ T cells

The TCRs hypervariable complementarity-determining region 3 (CDR3)α and CDR3β regions often mediate the fine pHLA class I specificity. We therefore used TCRdist to dissect the CDR3αβ clonotypic signatures by establishing their amino acid motif similarities to highlight key conserved residues that drive IBV epitope-specific TCR recognition (Fig. 9g). We identified a single TRAV1-2-TRAJ36-associated CDR3α-"CAVREGTG" motif (CDR3α chi-squared (chi-sq) 182.2) and a weaker TRBV5-4-associated CDR3β-"SDRExGG"-motif (chi-sq 90.2) for B7/NP₃₀⁺CD8⁺ TCRs. Neither motif was among the high frequency clonotypes, as their prominence was mainly driven by singletons (Supplementary Tables 5 and 6). For B8/NP₃₀⁺ CD8⁺ TCRs, only a strong TRBV19-TRBJ1-1-associated CDR3β-"ASSxxPDMN" motif was identified (chi-sq 552.2), which was present in high frequency clonotypes in 3 donors or singletons in one other donor (Supplementary Tables 5 and 7). B7/NS1₁₉₆₋₂₀₆-specific CDR3α motifs were dominated by a TRAV38/DV8-associated CDR3α-"YRSAxxYxLIW/F" motif (chi-sq 132.2), highly prevalent clonotypes in 1/6 donors and singletons in 2/6 donors, while a strong TRBV27-associated CDR3β-"CASSLYx" motif (chi-sq 306.2) was detected in highly prevalent clonotypes in 2/6 donors and singletons in 4/6 donors (Supplementary Tables 5 and 8). B8/NP₉₂⁺CD8⁺ TCRs displayed a TRBJ2-3-associated CDR3β-"xRxDTQ" motif, identified among high prevalent clonotypes and singletons (Supplementary Tables 5 and 9). No B8/NP₉₂⁺CD8⁺ CDR3α motifs were identified.

Overall, CDR3αβ motifs within HLA-B*07:02 and -B*08:02-restricted CD8⁺ T cell populations are distinct and present in either high prevalence clonotypes or singletons.

## Discussion

Despite the burden of IBV[2], only 18 IBV CD8⁺ T cell epitopes have been reported to date[24–29]. A broader array of highly conserved IBV CD8⁺ T cell epitopes is needed if we are to elicit effective CD8⁺ T cell immunity via a rational design of the universal cross-reactive IBV vaccine. Using our immunopeptidomics platform and in vitro peptide stimulation analyses, provide 209 IBV peptides, including

alternative reading frame and >10 amino acid peptides, for future investigation. Overall, we identified 9 IBV CD8[+] T cell peptides restricted to the prominent HLA-B*07:02 (B7/NP$_{30-38}$, B7/NP$_{233-244}$, B7/NS1$_{196-206}$, B7/HA$_{255-263}$), HLA-B*08:01 (B8/NP$_{92-99}$, B8/NP$_{479-487}$) and HLA-B*35:01 (B35/NS1$_{260-268}$, B35/HA$_{231-239}$, B35/PB$_{66-74}$) alleles. Additionally, we confirmed B8/NP$_{30-38}$ as an HLA-B*08:01-restricted epitope. Our study thus increased the overall number of known IBV CD8[+] T cell epitopes by 50% and covers 2 additional prominent HLA allomorphs, HLA-B*07:02 and HLA-B*35:01. Ex vivo phenotypic analyses revealed predominant T$_{cm}$ memory phenotype in IBV-specific tetramer[+]CD8[+] T cells within PBMCs, and T$_{rm}$ memory phenotype in tissues, especially in lungs, indicating establishment of IBV-specific T$_{rm}$ populations during past IBV infections. The frequencies of IBV-specific CD8[+] T cells peaked in adults, and although T$_{cm}$ populations peaked in elderly, there was a sharp decline in their naïve IBV-specific T cell compartment. Furthermore, our analyses of IBV-specific CD8[+] TCRαβ repertoires revealed high levels of TCR diversity within and between donors directed towards 4 IBV epitopes (B7/NP$_{30-38}$, B7/ NS1$_{196-206}$, B8/NP$_{30-38}$ and B8/NP$_{92-99}$).

Current inactivated influenza vaccines elicit substantial neutralizing antibodies against the vaccine strains but provide limited protection when the vaccine strains do not antigenically match the circulating strains. A key feature of CD8[+] T cells is their ability to recognize peptides derived from conserved internal influenza proteins driving interest in T cell-based universal cross-reactive vaccine strategies[24]. All epitopes identified in this study were highly conserved among known B/Victoria and B/Yamagata influenza B viruses (>99%), except NP$_{233-244}$ which was 91.9% conserved. These IBV-specific CD8[+] T cells are therefore likely to contribute to robust cross-reactivity to both IBV lineages[20], which makes them an important asset for future T cell-based influenza vaccines. For almost four decades, two influenza B lineages, B/Victoria and B/Yamagata, have been co-circulating during influenza seasons[2,54]. The B/Yamagata lineage has not been detected since 2020[12,55] due to the strict COVID-19 restrictions affecting circulation of other respiratory viruses. However, B/Yamagata viruses might still circulate locally at low levels in some parts of the world with potential to resurface globally in the future. Establishment of IBV-specific CD8[+] T cell responses towards highly conserved epitopes would provide some level of protection against IBV viruses of both B/Victoria and B/Yamagata influenza B lineages.

Previous studies established that age-related immunological changes, including those within the CD8[+] T cell compartment, may explain differences in disease severity across the human lifespan[41,56]. Indeed, IAV, IBV, SARS-CoV-2 and RSV have all demonstrated age-related differences in disease severity across human life. While RSV is most severe in young children[57], IAV causes severe disease in young children and elderly individuals[6] and COVID-19 particularly impacted the elderly[58]. However, IBVs cause severe infections in school-aged children[5,7,8]. As age-related changes in IAV[39,41] and SARS-CoV-2[59,60]-specific CD8[+] T cell populations have been previously detected, we hypothesized that age-related differences in IBV-specific CD8[+] T cells may explain the susceptibility of older children to IBV. We found that IBV-specific CD8[+] T cell populations peaked in adults and declined in the elderly, similar to what we observed in IAV A2/M1$_{58}$-specific CD8[+] T cells[38]. However, in contrast to IAV-specific CD8[+] T cells, the observed decline in IBV-specific CD8[+] T cells was associated with reduced frequencies of naïve IBV-specific CD8[+] T cells and further increasing T$_{cm}$ populations. The maintenance of high T$_{cm}$ frequencies directed against conserved IBV epitopes may provide substantial protection against severe IBV disease at older age, due to a high level of conservation within the immunogenic peptides across IBV strains. In contrast, children's IBV-specific CD8[+] T cells expressed a mixed naïve/ memory phenotype, which may be related to the number of exposures. A similar trend was observed for IAV A2/M1$_{58}$-specific CD8[+] T cells and may be attributed to the accumulation of repeated IAV

exposures throughout childhood[41]. This, however, does not explain why older children are at higher risk for severe IBV infections. One important difference between IAV and IBV viruses is that the first IBV exposure often occurs at a later age[61,62]. The SARS-CoV-2 pandemic demonstrated that young children benefitted from a strong innate immune response[63,64], while dampening the SARS-CoV-2-specific CD8[+] T cell response[59,60]. It will therefore be interesting to understand whether IBV-infected young versus older children mount different immune responses upon their first IBV encounter, resulting in either protective or detrimental disease outcomes.

We identified a single peptide, NP$_{30-38}$, is restricted by two HLA allomorphs, HLA-B*07:02 and HLA-B*08:01. A few examples of peptides binding multiple HLAs (named peptide promiscuity) have been previously described[65-68]. Three notable examples include (1) IAV M1$_{58-66}$ peptide restricted by HLA-A*02:01[69] and HLA-C*08:01[68], (2) IAV NP$_{418-426}$ peptide presented by HLA-B*07:02, HLA-B*35:01 and HLA-B*35:03[65,67], and (3) human immunodeficiency virus (HIV)-derived Pol$_{448-456}$ peptide, presented by HLA-B*35:01, HLA-B*51:01, HLA-B*53:01, and HLA-B*07:02[66]. Strikingly, the same CD8[+] TCR, Vα12.1/Vβ5.6, can recognize the Pol$_{448-456}$ peptide when presented on these HLA-I alleles. In contrast, the TCR repertoires of B7/NP$_{30}$[+]CD8[+] and B8/NP$_{30}$[+]CD8[+] T cells do not overlap. Distinct TCR repertoires suggest structural differences between HLA-B*07:02 or HLA-B*08:01 when presenting NP$_{30-38}$. Indeed, even subtle structural variations in pHLA conformation can have an impact on the TCR repertoire[70,71].

In addition to B7/NP$_{30-38}$- and B8/NP$_{30-38}$-specific CD8[+] TCR repertoires, we also defined B7/NS1$_{196}$[+]CD8[+] and B8/NP$_{92}$[+]CD8[+] TCR repertoires. At 11 amino acids long, the NS1$_{196-206}$ peptide has a length that surpasses the typical 8-10 amino acid length of HLA-I ligands[47], leading to a bulging confirmation when presented on HLA-B*07:02 consistent with other long (≥11 amino acids) HLA-I ligands[32,50]. Stable bulging peptides are usually associated with highly restricted/clonal TCR repertoires[49], whereas structural flexibility of bulging peptides more often results in diverse TCR repertoires[50,72,73]. In the case of B7/NS1$_{196-206}$, we observed a diverse TCR repertoire, which may be reflective of the low mobility of the bulged section.

In summary, we identified 9 IBV epitopes and provided important insights in IBV-specific CD8[+] T cell immunity across different tissue compartments and the human lifespan. Our identified IBV epitopes expand the number of IBV epitopes by 50%, which is of great importance for T cell based influenza vaccines, which provide an important layer of protection against severe IBV disease. We also defined age-related decrease in the frequencies of IBV-specific CD8[+] T cells and accumulation of T$_{cm}$ with age, which resembles our findings in IAV A2/M1$_{58}$[+]CD8[+] T cells[41].

## Methods
### Study participants and ethics statement
Adult and elderly participants were recruited via the University of Melbourne (UoM; Melbourne, Australia), Deepdene Medical Clinic (DMC; Deepdene, Australia) and the Australian Red Cross Lifeblood (ARCL; Melbourne, Australia). Children peripheral blood mononuclear cells (PBMCs), and children and adult tonsils were recruited via the Launceston General Hospital (Tasmania, Australia) from healthy individuals undergoing tonsillectomy. Lung samples were obtained from the Alfred Hospital's Lung Tissue Biobank. Spleen samples were obtained from DonateLife Victoria. All participants or their guardians provided informed written consent. HLA class I molecular genotyping was performed from genomic DNA by the ARCL. HLA profiles were used to select donors expressing HLA-B*07:02, HLA-B*08:01 and/or HLA-B35*01. PBMCs were isolated using Ficoll-Paque (GE Health-Care) gradient centrifugation, and cryopreserved in liquid N$_2$ until required. Tissues were processed and cryopreserved in liquid N$_2$ as described[17]. Experiments were conducted in accordance with the Declaration of Helsinki Principles and according to the Australian National Health and

Medical Research Council Code of Practice. The study was approved by the Human Research Ethics Committee (HREC) of the University of Melbourne Experiments (Ethics ID 13344, 24567), Australian Red Cross Lifeblood (ID 2015#8) and Tasmanian Health and Medical HREC (ID H0017479).

## Cell lines and reagents

Parental C1R[74] (a gift from Weisan Chen, La Trobe University), C1R.B*07:02[75], C1R.B*08:01[74,76] (a gift from Zhenjun Chen, University of Melbourne) and C1R.B*35:01[74] (a gift from Anthony Purcell, Monash University) cell lines were maintained in RF10 medium (RPMI1640 medium (Invitrogen) supplemented with 2mM L-glutamine (Gibco), 1 mM MEM sodium pyruvate (Gibco), 100 μM MEM non-essential amino acids (Gibco), 5 mM HEPES buffer solution (Gibco), 55 μM 2-mercaptoethanol (Gibco), 100 U/ml penicillin (Gibco), 100 μg/ml streptomycin (Gibco) and 10% fetal bovine serum (Gibco). Influenza B (IBV, B/Malaysia/2506/04) virus was grown in the allantoic cavity of day 10-old embryonated chicken eggs for 3 days at 35 °C and viral titers were determined by plaque assay on MDCK cells (American Type Culture Collection). Peptides were purchased from GenScript or Mimotopes and reconstituted to 1 mM with dimethyl sulfoxide (DMSO; Sigma-Aldrich).

## Previously discovered IBV epitopes

The Immune Epitope Database (IEDB)[30] was used to search for known IBV CD8+ T cell epitopes. The parameters for the search were for linear peptides, human hosts, T cell assays and MHC Class I restriction. References for each peptide were analyzed to ensure that epitopes were confirmed ex vivo, in vitro or in vivo, excluding peptides not claimed as being epitopes by the respective studies. Peptides with single amino acid substitutions or overlapping sequences were counted as a single epitope.

## LC-MS/MS analysis of immunopeptidome

C1R were infected at MOI5 for 12hrs and snap frozen as previously[26]. HLA class I complexes were isolated as described[31]. Briefly, C1R cell pellets were disrupted by cryogenic milling (Retsch Mixer Mill MM 400) and lysed with 0.5% IGEPAL CA-630, 50 mM Tris-HCl pH 8.0, 150 mM NaCl and protease inhibitors (cOmplete Protease Inhibitor Cocktail Tablet; Roche Molecular Biochemicals) for 1 h at 4 °C with rolling. Lysates were cleared by ultracentrifugation and the HLA class I complexes isolated by immunoaffinity purification using pan-HLA class I antibody (W6/32) bound to protein-A-agarose. Peptide-HLA were dissociated with 10% acetic acid and fractionated by reversed-phase high-performance liquid chromatography. Fractions containing peptides were combined into 9–10 pools, vacuum-concentrated, then reconstituted in 15 μl 0.1% formic acid in Optima™ LC-MS water with indexed retention time (iRT) peptides[77]. Peptides were analyzed by LC-MS/MS on a Q-Exactive Plus Hybrid Quadrupole Orbitrap (Thermo Fisher Scientific) as described[24,25].

## Bioinformatic analysis of LC-MS/MS data

For each cell line, spectra were searched in an individual project using PEAKXPRO v10.6 (Bioinformatics Solutions Inc) database search. Spectra were searched against the reviewed human proteome (downloaded from UniProt March 2022) with the proteome and 6 frame translated nucleotide sequence of influenza B/Malaysia/2506/04 virus appended, and the iRT peptides. False Discovery Rate (FDR) analysis was performed by decoy-fusion. The following settings were applied: Instrument: Orbitrap (Orbi-Orbi), Fragment: HCD, Acquisition: DDA, Associate Chimera Scan: Enabled, Parent Mass Error Tolerance: 10.0 ppm, Fragment Mass Error Tolerance: 0.02 Da, Precursor Mass Search Type: Monoisotopic, Enzyme: None, Digest Mode: Unspecific, Variable Modifications: Oxidation (M) +15.99 Da,

Cysteinylation +119.00 Da and Deamidation (NQ) +0.98 Da, Max Variable PTM Per Peptide: 3. Data from previous analyses of presentation by endogenous HLA of C1R cells (endogenous HLA data sets: A/X31[26], B/Malaysia/2506/04[24]) were searched with the same settings.

To characterize the binding motifs, a peptide-level FDR cut-off of 1% was applied. Analyses were performed with non-redundant human peptide sequences. For all HLA transfectants, peptide sequences were excluded if identified in immunoprecipitations of endogenous HLA of the C1R derivates using DT9 (isolates HLA-C*04:01) or a mix of LB3.1, SPV-L3 and B721 (isolate HLA-DR, -DQ and -DP) in the endogenous HLA data sets[24,26]. The C1R.B*08:01 dataset was further filtered of peptide sequences isolated using W6/32 in the endogenous HLA data sets to remove endogenous HLA-B*35:03 binders. This was not performed for the C1R.B*07:02 and C1R.B*35:01 datasets to avoid elimination of binders shared with the transfectant HLA given HLA-B*07:02, HLA-B*35:01 and HLA-B*35:03 all favor Proline at P2. Gibbs cluster analysis was performed on the unfiltered human 8-13mers for each data set using GibbsCluster2.0[78,79] (recommended configuration MHC class I ligands of length 8-13), and those peptide sequences that clustered with experimentally identified HLA-C*04:01 ligands were also excluded (Supplementary Fig. 6). Sequence Logos were generated with Seq2-Logo2.0 using default settings[80] and graphs were generated using GraphPad Prism 9.5 for Windows (GraphPad Software, San Diego, California USA, www.graphpad.com).

To characterize influenza peptides, a peptide FDR cut-off of 5% was set. Peptide assignments mapping to the influenza proteome or 6 frame genome translation were excluded if identified in both uninfected and infected samples with similar spectra and retention times. For peptides mapping to alternative reading frames, naming and positional numbering was according to the frame starting at the start of the viral UTR, with stop codons counted as an amino acid. Predicted binding affinities of peptides between 8–14 amino acids in length to the transfected HLA (HLA-B*07:02, HLA-B*08:01 or HLA-B*35:01) and endogenous HLA (HLA-B*35:03 and HLA-C*04:01) were calculated using NetMHC4.0[81,82] using the native peptide sequence (unless otherwise stated), and assigned as strong binders (SB) and weak binders (WB) if the % rank was below 0.5 and 2, respectively. Peptides were considered potential binders of HLA-B*08:01 if the % rank or nM Affinity was lower than that for B*35:03 and C*04:01 and below 20. For B*07:02 and B*35:01, peptides were considered potential binders if the % rank or nM Affinity was lower than that for C*04:01 and below 20. For B*07:02 and B*35:01 predicted binding to B*35:03 was not considered to reduce likelihood of binding the transfected HLA due to similar preference for Proline at P2. As with human-derived peptides, sequences seen in endogenous HLA-class II data sets[24] were excluded in figures, however these identifications are noted in Supplementary Data 1.

Datasets have been deposited to the ProteomeXchange Consortium via the PRIDE[83] partner repository with the dataset identifier PXD045000 and 10.6019/PXD045000.

## Expansion of IBV-specific CD8+ T cells from human PBMCs

PBMCs ($3–5 \times 10^6$) were used to expand antigen-specific CD8+ T cells[24–26]. Briefly, PBMCs were thawed in RF10 supplemented with 2 μg/ml deoxyribonuclease I (DNase; Sigma-Aldrich) and washed in serum-free RPMI. One third of PBMCs were pulsed for 1 h with IBV peptide pools (7 peptide pools ranging between 8 and 22 peptides per pool; Supplementary Table 1) in serum-free RPMI at 10 μM at 37°C. Pulsed cells were washed twice with serum-free RPMI, resuspended in RF10 and mixed with the remaining autologous PBMCs (also resuspended in RF10). Cultures were incubated at 37 °C/5% $CO_2$ for 10–12 days. IL-2 (Roche Diagnostics) was added on day 4 to a final concentration of 20 U/ml.

### Restimulation and intracellular cytokine staining of CD8+ T cells

Expanded IBV-specific CD8+ T cells were harvested on day 10 and restimulated with the cognate peptide pool or PMA/Ionomycin and DMSO as controls for 5 h in the presence of Brefeldin A (Golgi Plug, BD), Monensin (Golgi Stop, BD) and anti-CD107a-FITC (1:200, Invitrogen #53-1079-42) at 37 °C/5% $CO_2$. After restimulation, cells were surface stained on ice for 30 min with Live/Dead near infrared (NIR) (1:800, Invitrogen #L34976), anti-CD8-PerCP-Cy5.5 (1:00, BD Pharmingen #565310), anti-CD4-PE (1:50, BD Pharmingen #555347) or anti-CD4-BV650 (1:200, BD Horizon #563875) and anti-CD3-PE-Cy7 (1:50, BD Pharmingen #563423), fixed and permeabilised with BD Cytofix/cytoperm and intracellularly stained on ice for 30 min with anti-IFN-γ-V450 (1:100, BD Pharmingen #560371), anti-MIP-1β-APC (1:40, BD Pharmingen #560686) or anti-MIP-1β-PE (1:50, BD Pharmingen #550078), anti-TNFα-AF700 (1:50, BD Pharmingen #557996), acquired on a LSRII Fortessa (BD Biosciences) and subsequently analyzed with FlowJo v10.8.1 (FlowJo, LLC). On day (d) 11 or d12, expanded cell populations responding to immunogenic peptide pools were harvested and restimulated for 5 h with either individual IBV peptides at 10 μM, the peptide pool, PMA/Ionomycin or DMSO and subsequently stained as described above. Antigen-specific CD8+ T cells expanded for 11-12 days with pools consisting of immunogenic peptides only (Supplementary Table 3) were restimulated for 5 h with C1R.B*07:01, C1R.B*08:01, C1R.B*35:01s or parental C1Rs pulsed with IBV peptides at 10 μM in the presence of Brefeldin A (BD Golgi Plug), Monensin (BD Golgi Stop) and anti-CD107a FITC (Invitrogen) at 37 °C/5% $CO_2$ and stained as above.

### Conservation Influenza B virus-derived peptides

Conservation of IBV peptides was performed as described[25,84]. Briefly, 16,645 full-length HA (1940 to 9 September 2022), 11,111 full-length NP (1940 to 26 November 2021) and 12,987 full-length NS1 (1940 to 23 March 2023) amino acid (aa) sequences from human isolates were downloaded from the influenza virus resource database of the National Center for Biotechnology Information database (NCBI; http://www.ncbi.nlm.nih.gov/genomes/FLU), accession numbers can be found in Supplementary Data 2. Sequences with large deletions were excluded (Bioedit v7.2.5) from further analysis. Overall 11 HA, 10 NP and 2 NS1 aa sequences were omitted. Conservation of each aa residue of the relevant peptides was established using UGENE v40.1 (http://ugene.unipro.ru/, Unipro). The overall conservation of the peptide sequence calculated as the average conservation of the combined aa positions in the peptide.

### HLA/peptide tetramer staining of expanded IBV-specific CD8+ T cell lines

pHLA class-I (pHLA-I) tetramers were generated by refolding each peptide with its restricted HLA α-heavy chain-BirA and β2-microglobulin[53,85,86] before 8:1 conjugation with PE- or APC-streptavidin (BD, cat #554061 and #554067 respectively) to generate tetramers; B7/NP30, B7/NS1196, B8/NP30, B8/NP92, B8/NP479, B35/HA231 and B35/NS1260. To test newly generated tetramers, 0.2–0.4 × 10^6 frozen IBV pool-specific CD8+ T cells were thawed in RF10 supplemented with 2 μg/ml DNAse. Cells were incubated with anti-human FcR block (Miltenyi Biotec) for 15 min and stained with tetramer at room temperature for 1 h before surface staining on ice for 30 min with anti-CD3-BV510 (1:200, BioLegend #317332), anti-CD4-BV650 (1:200, BD Horizon #563875), Live/Dead NIR (1:800, Invitrogen #L34976) and anti-CD8-PerCP-Cy5.5 (1:50, BD Pharmingen #565310). Cells were then fixed with 1% paraformaldehyde (PFA) (Electron Microscopy Sciences) for 20 min on ice, acquired on an LSRII Fortessa and analyzed with FlowJo v10.8.1.

### Ex vivo tetramer associated magnetic enrichment (TAME) of epitope-specific CD8+ T cells

PBMCs, spleen, lymph nodes or tonsils (10–50 × 10^6) were thawed in RF10 supplemented with 2 μg/ml DNAse or 50U/ml Benzonase (Novagen Merck). Spleen samples were passed through LS columns after thawing to prevent contamination with red pulp, as described[87]. Tetramer-specific CD8+ T cells were enriched using magnetic separation[32]. Briefly, cells were stained with the B7/NP30, B7/NS1196, B8/NP30, B8/NP92, B8/NP479, B35/HA231 or B35/NS1260 tetramers at room temperature for 1 h in MACS buffer (PBS with 0.5% BSA and 2 mM EDTA) and incubated with anti-PE and/or anti-APC microbeads (Miltenyi Biotec) followed by magnetic separation. PBMC samples were then stained with anti-CD71-BV421 (1:50, BD Horizon #562995), anti-CD3-BV510 (1:200, BioLegend #317332), anti-HLA-DR-BV605 (1:100, BioLegend #307640), anti-CD4-BV650 (1:200, BD Horizon #563875), anti-CD27-BV711 (1:200, BD Horizon #563167), anti-CD38-BV786 (1:100, BD Horizon #563964), anti-CCR7-AF700 (1:50, BD Pharmingen #561143), anti-CD14-APC-H7 (1:100, BD Pharmingen #560180), anti-CD19-APC-H (1:100, BD Pharmingen #560177 or #560252), anti-Live/Dead NIR (1:800, Invitrogen #L34976), anti-CD45RA-FITC (1:200, BD Pharmingen #555488), anti-CD8-PerCP-Cy5.5 (1:50, BD Pharmingen #565310), anti-CD95-PE-CF594 (1:100, BD Horizon #562395) and anti-PD-1-PE-Cy7 (1:50, BD Pharmingen #561272). Tissue samples were stained with the panel above including anti-CD103-BUV395 (1:50, BD Horizon #564346) and anti-CD69-BV421 (BioLegend #310930). Cells were either resuspended in MACS for single cell index sorting using the FACSAriaIII (BD) or fixed with 1% PFA before acquiring on an LSRII Fortessa and subsequently analyzed with FlowJo v10.8.1.

### Tetramer staining within human lungs

Lung samples (7−20 × 10^6 cells) were thawed, resuspended in MACS buffer and run through a 100 μm filter followed by a 30 μm filter. Samples were incubated with anti-human FcR block for 15 min, stained with pMHC-I for 1 h at room temperature, surface stained with the tissue TAME panel on ice for 30 min and fixed in 1% PFA on ice for 20 min. Samples were acquired on an LSRII Fortessa and analyzed with FlowJo v10.8.1 (FlowJo, LLC).

### Single-cell multiplex paired TCRαβ sequencing

Enriched tetramer+ CD8+ T cells were single cell index sorted into chilled 96 well twin.tec PCR plates (Eppendorf) and immediately stored at −80 °C. Multiplex-nested RT-PCR of paired CDR3α and CDR3β regions was performed as described[32,88]. Primers used can be found in Supplementary Table 10. Sequences were analyzed with FinchTV. V-J regions were identified with IMGT (www.imgt.org/IMGT_vquest). The TCRdist analytical pipeline[89] was used to parse TCR sequences. Clonotypes were defined as single-cell TCRαβ pairs that exhibit the same V, J and CDR3 regions. The motifs are computed by the TCRdist algorithm and are reflective of CDR3 sequences of variable lengths. Circos plots were generated with the circlize package[90] in R v4.1.3 (The Comprehensive R Archive Network (CRAN)).

### CDR3αβ motif analysis

TCRdist[88] was used to identify CDR3 motifs. In brief, TCRdist performs a statistical analysis of overrepresented CDR3 sequence motifs, and takes in account the underlying sequence biases introduced by the rearrangement process. Furthermore, TCRdist uses a recursive search algorithm to identify sequence patterns that occur significantly more often in the observed receptors than in two V and J gene matched background sets of receptor sequences.

### Protein production and purification

Peptides used in this study were >85% pure and purchased from GenScript. Human β2m micro-globulin and HLA-B*07:02, B*08:01 and B*35:01 heavy chains with C-terminal Bir-A tag were expressed as inclusion bodies in *E.coli BL21 (DE3)*. The HLAs were refolded and purified as described[91]. Briefly, ~60 mg of HLA-B*07:02BirA, -B*08:01BirA and -B*35:01BirA heavy chain, 20 mg of human β2m and 6 mg of peptide (Table 1) incubated in refolding buffer containing 3 M urea, 100Mm

Tris (pH-8.0), 400 mM Arginine, 2 mM EDTA, 0.1 mM PMSF, one protease inhibitor tablet (Roche), and 0.5 mM oxidized and 5 mM reduced Glutathione. The refolded HLAs-peptide complex was purified from DEAE anion exchange and Size exclusion chromatography. HLA-B*07:02$_{BirA}$, HLA-B*0801$_{BirA}$ and B*35:01$_{BirA}$ refolded with different peptides are biotinylated with Bir-A ligase which was further used for tetramer staining.

## Crystallization, data collection and structure determination

Crystals for HLA-B*07:02-NS1$_{196-206}$ (HPNGYKSLSTL) and HLA-B*07:02-NP$_{30-38}$ (RPIIRPATL) binary complex were crystallized using vapor diffusion method at 20 °C in 1:1 drop ratio of protein and reservoir. The binary complex crystals of B702$_{NS196-206}$ were obtained at 20 mg/ml protein concentration using mother liquor 0.2 M Sodium sulfate, 20 %w/v PEG 3350. While crystals for B*07:02-NP$_{30-38}$ were grown at 15 mg/ml protein concentration using mother liquor 0.2 M NH$_4$NO$_3$, 20 %w/v PEG 3350. Crystals of both binary complexes were gradually transferred into mother liquor supplemented with 20% Ethylene Glycol and subsequently flash-frozen in liquid nitrogen. Crystals were diffracted at MX-1 and MX-2 beamlines of Australian Synchrotron at Dectris Eiger detector respectively. Data were processed using XDS[92] and Aimless from the CCP4 program suite[93]. The structures were solved via molecular replacement in Phaser[94] and structure models were built via iterative rounds of model building in Coot[95] and restrained refinement in Phenix[96]. Structural overlays of HLA-B*08:01 and HLA-B*35:01 onto the epitope conformation of HLA-B*07:02-NP$_{30-38}$ were created using previously determined versions of these MHC platforms. Structures were deposited at www.rcsb.org with codes 8TUH and 8TUB.

## Statistical analysis

Statistical analysis was performed using GraphPad Prism v9.5.1 (GraphPad). Statistical significance was determined using Kruskal–Wallis test with Dunn's correction for multiple tests. For paired spleen and lymph node samples, Wilcoxon tests with no correction for multiple comparisons were used. Differences were considered significant at $p$ value < 0.05. No statistical methods were used to pre-determine sample sizes but our sample sizes were similar those reported in previous publications. Experiments were not randomized and Investigators were not blinded to allocation during experiments and outcome assessment. Samples with <10 tetramer$^+$ T cells counted were excluded for phenotypic analyses.

## Reporting summary

Further information on research design is available in the Nature Portfolio Reporting Summary linked to this article.

## Data availability

TCR sequence data (Supplementary Tables 6–9, Source data) has been deposited in Mendeley [https://doi.org/10.17632/wg7swwf6jr.3] and VDJdb (https://vdjdb.cdr3.net). Mass spectrometry proteomics data have been deposited in the ProteomeXchange Consortium via the PRIDE[83] partner repository with the dataset identifier PXD045000. Crystal structures have been deposited into the RSCB protein data bank ([https://www.rcsb.org/structure/8TUH]; HLA-B*07:02-NP$_{30-38}$) and ([https://www.rcsb.org/structure/8TUB]; HLA-B*07:02-NS1$_{196-206}$). The raw data generated in this study are provided in the Source Data file. Any additional information needed to reanalyze the data in this paper is available from the corresponding author upon reasonable request. Source data are provided with this paper.

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

## Acknowledgements

We thank the participating donors involved in the study as well as Bernadette McCudden for their medical support. We thank the Melbourne Cytometry Platform for the technical support and assistance. Computational resources for proteomics analysis were supported by the R@CMon/Monash Node of the NeCTAR Research Cloud, an initiative of the Australian Government's Super Science Scheme and the Education Investment Fund. The authors acknowledge the provision of instrumentation, training and technical support by the Monash Proteomics and Metabolomics Platform and thank Dr Marios Koutsakos for discussions. This work was supported by the Clifford Craig Foundation Project Grant to K.F. and K.K. (#186), the Research Grants Council of the Hong Kong Special Administrative Region, China (#T11-712/19-N) to K.K. and the NHMRC Leadership Investigator Grant to K.K. (#1173871) and Michelson Medical Research Foundation and Human Vaccines Project Michelson Prize for Human Immunology and Vaccine Research to P.T.I. P.T.I. was supported by Monash University Faculty of Medicine, Nursing and Health Sciences Senior Postdoctoral Fellowship (2020). C.S. is a recipient of the Australian Government Research Training Program Stipend. C.E.S. is supported by the ARC-DECRA Fellowship (#DE220100185) and the University of Melbourne Establishment Grant. T.H.O.N. is supported by the NHMRC Emerging Leadership Level 1 Investigator Grant (#1194036), J.R. is supported by an NHMRC Investigator Grant. A.W.P. is supported by a NHMRC Investigator grant (# 2016596). The project was support by a NIAID UO1 grant 1U01AI144616-01 "Dissection of Influenza Vaccination and Infection for Childhood Immunity" (DIVINCI) and NIH contract CIVR-HRP (HHS-NIH-NIAID-BAA2018) to K.K.

## Author contributions

K.K. led the study. K.K., C.E.S., A.W.P. and J.R. supervised the study. T.M., P.T.I., P.C., J.P., D.R.L., N.A.M., J.R., A.W.P., C.E.S. and K.K. designed the experiments. T.M., P.T.I., P.C., C.S., L.C.R., G.K., Z.H., L.F.A., T.H.O.N. and C.E.S. performed experiments. T.M., P.T.I., P.C., H.A.M., C.S. and C.E.S. analyzed the data. S.R., J.C., K.F. and L.M.W. provided reagents or samples. T.M., P.T.I., P.C., C.S., L.C.R., J.P., D.R.L., N.A.M., J.R., A.W.P., C.E.S. and K.K. provided intellectual input into study design and data interpretation. T.M., P.T.I., P.C., C.E.S. and K.K. wrote the manuscript. All authors reviewed and approved the manuscript.

## Competing interests

S.R. is an employee of CSL, a company that produces influenza vaccines. A.W.P. is a scientific advisor for Bioinformatics Solutions Inc (Canada) and Grey Wolf Therapeutics (UK), a shareholder and scientific advisor for Evaxion Biotech (Denmark), and a co-founder of Resseptor Therapeutics (Australia). These interests had no role in the design of the study; in the collection, analyses, or interpretation of data; in the writing of the manuscript; or in the decision to publish the results. The remaining authors declare no competing interests.
