## [Peer Review File · Nature Communications]

Prominent CD8+ T-cell responses towards conserved influenza B virus epitopes across anatomical sites and ageREVIEWER COMMENTS

Reviewer #1 (Remarks to the Author):

This paper by Menon et al reported the identification of cross-reactive influenza virus B-specific T cells which are relatively less studied and required to understand the protective T cell immunity. They focused on the relatively common HLA-B allele, namely B*07:02, HLA-B*35:01 and HLA-B*08:01. In total, 9 epitopes were identified and validated by ex-vivo phenotyping of tetramer-positive cells in both blood and tissue, which were dominated by a central memory phenotype, not surprising but nice to see. Interestingly, they found an epitope in NP known to be presented by HLA-B0801 and could also be presented by HLA-B0702, however with distinct TCR repertoires. The structures of the two epitopes identified/presented by HLA-B0702 were also provided, which is very useful in understanding HLA-B0702 as a prominent HLA in presenting viral peptides in a potentially efficient way, as well as the structure differences between long vs short peptide presentation using two newly identified epitope peptides - both presented by HLA-B0702. Overall, I think this is a very important, solid and timely study. I only have the following few specific comments:

Is there any preference of this NP30 epitope presented by HLA-B7 or B8? From Figure 9a it seems like NP30 is preferentially presented by HLA-B*07:02 when both alleles are expressed. HLA-B7/NP30 tetramer staining has a clear population, whereas HLA-B8/NP30 staining is very weak with a smear staining, indicating some tetramer stained cells could be non-specific or with a very low affinity. This may explain the more diverse TCR usage for the B8/NP30-specific T cells, especially TCRs from B7/B8 double positive donors. It might be worth to show flow cytometry plots for a HLA-B8+ only donor; it would also be interesting to see if there are any phenotype difference between HLA-B7/NP30-specific T cells and HLA-B8/NP30-specific T cells overall including in B7 and B8 double positive donors and single positive donors?

Reviewer #2 (Remarks to the Author):

The manuscript in question is based on sound methods and interpretations and the issues raised below would not have a major impact on the conclusions drawn. Here is a summary of my understanding of the work performed:

Influenza B infections account for a significant fraction of seasonal influenza infections, yet are understudied as compared to Influenza A. The lack of known epitopes prevents the design of vaccines that target both IVA and IVB. The authors first established a ground truth by observing the immunogenicity of several known epitopes or HLA-B*08:01 in healthy individuals expressing that allele. Novel candidate epitopes were identified through an established immunopeptidomic pipeline. Peptides were stratified according to confidence in identification and predicted binding affinities and selected for further analysis. Peptide pools were tested for immunogenicity and further deconvoluted to determine the responses of individual peptides. HLA restrictions were confirmed through IFN γ and TNF α assays of pool-specific CD8 $^+$ T cells to peptide-pulsed C1R cells expressing the alleles of interest. The composition of the responding CD8 $^+$ T cells was determined through tetramer assays, with the majority of cells falling into the Tcm class. When stratified by age, it was determined that IBV-specific CD8 $^+$ T cell populations decline with increasing age. The frequency of IBV-specific CD8 $^+$ T cells was assessed and noted to be higher in lymph node and spleen than

in PBMCs, although the highest frequency of the Trm phenotype was in the lung. The crystal structure of the 11mer NS1 peptide in complex with HLA-B*07:02 was solved to reveal a bulge in the epitope conformation. The crystal structure of an NP epitope in complex with HLA-B*07:02 was obtained and extrapolated to HLA-B*08:01 and HLA-B*35:01 to explain the lack of responses detected against the latter. An analysis of TCR repertoires indicated relatively high diversity between individuals for each peptide:MHC as well as differential gene usage associated with each epitope. Motif analysis yielded divergent CDR3a and b motifs among the 4 peptide:MHC complexes studied, mainly driven by lower frequency clonotypes.

Moderate concerns:

* In several places throughout the manuscript, the authors state that only 18 CD8+ IBV-specific epitopes have been reported in the literature. However, a search with these parameters at the IEDB (www.iedb.org) yields 39. Although this doesn't fundamentally change the outcome of the research, the significance may be overstated as a 50% increase in the number of known epitopes versus 20%. Can the authors help to clarify the discrepancy?

* MHC:peptide binding affinity predictions were performed with NetMHC4.0, which was a state-of-the-art method in 2015, but has since been surpassed in performance by other methods, including NetMHCPan 3.0 and above. Would the outcome, in terms of peptides selected for further analysis, have changed if NetMHCPan 4.0 or higher was used for the predictions?

* Methods, lines 739-745: Different thresholds, in terms of percentile rank and binding affinities, were applied to several alleles. In some cases, the thresholds seem interdependent. I could not find any justification for this approach in the manuscript as opposed to a uniform threshold across alleles.

* Data availability: I did not see an SRA identifier for the TCR sequencing data. Will this be made publicly available at SRA or another resource?

Minor concerns/questions:

* Table 1: The HA-255 epitope appears to be missing from this table.

* Figure 6: The authors conclusion that IBV-specific T cell responses decline with age is mostly supported by the data presented in Figures 6b and c. However, the relationship appears to be clearer in the donors above 18 years of age. If the responses were averaged among the 3 groups, it looks as though the responses in the 'children' group would be similar to or lower than those in the 'adults' group in many cases. Have the data been grouped in this manner to evaluate whether the relatively low number of subjects in the children group are adding noise to the signal?

* The authors selected B/Malaysia/2506/04 as the representative strain, but did not provide rationale. It may be widely known in the field, but I'm not an IBV expert and wonder how this strain was selected.

* Conservation analysis: Are there more specifics on the query parameters used to download the IBV sequences from NCBI, or can you provide a list of identifiers? Was there

any effort to remove redundancy and evaluate conservation among distinct sequences?

* Figure 9d: I could not find information on the methods used for motif identification for the CDR3a and b regions. As GibbsCluster was mentioned in the epitope analysis, it is assumed that this tool was also used here. Can you clarify? Regardless, is there a reason one might expect to identify a fixed-length motif, given the length variability of CDR3 sequences?

* It might be worth including metrics regarding the conservation of the novel IBV epitopes among IAV strains to speculate about the potential for cross-protective responses.

Reviewer #3 (Remarks to the Author):

The manuscript by Menon et al. explores influenza B (IBV) derived peptides presented by HLA Class I. The premise of the study is that IBV has not been well characterized for class I HLA restricted CD8+ T cell targets that potentially represent promising cellular vaccine candidates. The authors examine the global frequency of class I HLA alleles in their cohort and test the immunogenicity of IBV peptides in HLA-B*07:02, HLA-B*08:01, and HLA-B*35:01. Immunopeptidomics identifies 209 IBV peptides presented by HLA-B*07:02, HLA-B*08:01, and HLA-B*35:01. The authors condense these 209 peptides into seven pools for immunogenicity screening using PBMCs from healthy individuals with the aforementioned HLA-B alleles. The authors identify 9 novel IBV CD8+ T cell reactive peptides from NP, NS1, HA, and PB. Tetramer staining demonstrates TCM cells in the blood, TRM in the lungs, and that T cell phenotypes change with age. The presentation of NP30-38 by multiple HLA is then demonstrated with recognition by distinct TCR. The study then proceeds to structurally characterize NS1 196-206 and NP 30-38 in B*07:02. The stated impact is that considerably more IBV CD8+ T cell epitopes are now realized with a better understanding of CD8+ responses to these epitopes.

There are no apparent flaws or weaknesses with the experimental approaches. Indeed, the study is a experimental tour de force. The proteomics work is thorough, the immunogenicity testing that follows provides a complete hierarchy of T cell responsiveness/phenotype, and the tetramers facilitate the in-depth characterization of T cells restricted by these HLA/IBV complexes. At this point, however, the study loses momentum – the reader anticipates a continued distillation of the HLA/IBV complexes and their corresponding T cell responses into a mechanistic determination of T cells to specific HLA/IBV complexes that either do or do not protect from IBV infection – the stated objective of the study. While the experiments to characterize the HLA/IBV complexes and T cells that recognize these complexes in different tissues and individuals are done with the utmost precision, the data never stratify into protective or non-protective HLA/IBV complexes and T cells that correspond to protective immunity; no correlates of immune protection to infection from IBV emerge from the data. No mechanism of HLA/IBV restricted CD8+ T cell immune protection from IBV infection emerges from the study. It is therefore difficult for the reader to ascertain the relevance of the data in this report as it pertains to influenza immunity.

The latter part of the manuscript includes data pertaining to a peptide shared by both B*07:02 and B*08:01 that is 11 AA long and that bulges from the groove as demonstrated by

crystallization. How is this relevant to the main tenet of the manuscript? These data are quite interesting and worthy of further exploration, but they do not fit the narrative of the manuscript.

The authors make an argument in their discussion for their inclusion of a Yamagata strain-derived peptide in the study as being relevant due to the conserved nature of the peptide between it and the Victoria IBV strain. However, the language pertaining to a potential epidemic affecting millions due to waning nAb seems speculative.

We thank the Reviewers for their comments and insightful suggestions, which have allowed us to further improve the manuscript. We would also like to thank the Editors for the opportunity to resubmit the revised version to *Nature Communications*.

RESPONSES TO REVIEWER'S COMMENTS

We thank the Reviewers for their positive and encouraging comments: "Overall, I think this is a very important, solid and timely study." (Reviewer 1); "The manuscript in question is based on sound methods and interpretations..." (Reviewer 2); "There are no apparent flaws or weaknesses with the experimental approaches. Indeed, the study is a experimental tour de force. The proteomics work is thorough, the immunogenicity testing that follows provides a complete hierarchy of T cell responsiveness/phenotype, and the tetramers facilitate the in-depth characterization of T cells restricted by these HLA/IBV complexes." (Reviewer 3).

We respond to each Reviewer's questions and comments in a point-by-point form.

REVIEWER #1:

This paper by Menon et al reported the identification of cross-reactive influenza virus B-specific T cells which are relatively less studied and required to understand the protective T cell immunity. They focused on the relatively common HLA-B allele, namely B*07:02, HLA-B*35:01 and HLA-B*08:01. In total, 9 epitopes were identified and validated by ex-vivo phenotyping of tetramer-positive cells in both blood and tissue, which were dominated by a central memory phenotype, not surprising but nice to see. Interestingly, they found an epitope in NP known to be presented by HLA-B0801 and could also be presented by HLA-B0702, however with distinct TCR repertoires. The structures of the two epitopes identified/presented by HLA-B0702 were also provided, which is very useful in understanding HLA-B0702 as a prominent HLA in presenting viral peptides in a potentially efficient way, as well as the structure differences between long vs short peptide presentation using two newly identified epitope peptides - both presented by HLA-B0702. Overall, I think this is a very important, solid and timely study.

We thank the Reviewer for their positive feedback.

I only have the following few specific comments:

Is there any preference of this NP30 epitope presented by HLA-B7 or B8? From Figure 9a it seems like NP30 is preferentially presented by HLA-B*07:02 when both alleles are expressed. HLA-B7/NP30 tetramer staining has a clear population, whereas HLA-B8/NP30 staining is very weak with a smear staining, indicating some tetramer stained cells could be non-specific or with a very low affinity. This may explain the more diverse TCR usage for the B8/NP30-specific T cells, especially TCRs from B7/B8 double positive donors. It might be worth to show flow cytometry plots for a HLA-B8+only donor; it would also be interesting to see if there are any phenotype difference between HLA-B7/NP30-specific T cells and HLA-B8/NP30-specific T cells overall including in B7 and B8 double positive donors and single positive donors?

We thank the Reviewer for their suggestions. NetMHC4.0 analysis is in line with the Reviewer's hypothesis, as it reveals a stronger predicted binding affinity for NP₃₀₋₃₈ restricted to HLA-B*07:02 (10.3nM) than HLA-B*08:01 (47.2nM) (Supplementary Data 1), which may in turn lead to a

preferential B7/NP₃₀⁺ response (higher frequency). However, both are still classified as Strong Binders (SB).

In accordance with the Reviewer's suggestion, we have now included representative FACS plots for HLA-B*07:02 and -B*08:01 single and double positive donors in Fig. 9a (Rebuttal Fig. 1).

Rebuttal Fig. 1 (Fig. 9a in the revised version of the manuscript)

The difference in staining intensity likely results from the respective fluorophores used. PE is a much brighter fluorophore compared to APC and will therefore result in a better separation. Indeed, B8/NP₃₀ conjugated to PE provides a brighter staining in the same donor (Rebuttal Fig. 2).

Rebuttal Fig. 2 Same donor B8/NP₃₀-PE (left) and B8/NP₃₀-APC (right)

To test the Reviewer's hypothesis, we directly compared the *ex vivo* frequency of B7/NP₃₀ and B8/NP₃₀-specific CD8⁺ T cells in HLA-B*07:02 and -B*08:01 single and double positive donors. We did not identify any significant differences in overall frequencies, whether alleles were single or co-expressed (Rebuttal Fig. 3; Same as Supplementary Fig. 5a).

Rebuttal Fig. 3 (Supplementary Fig. 5a in the modified version of the manuscript)

By directly comparing the phenotypes of single- and double-positive donors, we observed prominent T_{cm} population for B7/NP₃₀ in HLA-B*07:02-expressing donors regardless of HLA-B*08:01 co-expression. A visually apparent but statistically indistinguishable trend for higher B8/NP₃₀⁺ T_{cm} frequencies was observed for HLA-B*08:01⁺ donors versus double-positive donors (Rebuttal Fig. 4; Supplementary Fig. 5b in the modified version of the manuscript) These results are in line with the Reviewer's hypothesis that the co-expression of HLA-B*07:02 may lead to a lower recruitment of B8/NP₃₀⁺ cells.

Rebuttal Fig. 4 (Supplementary Fig. 5b in the modified version of the manuscript)

In accordance with the Reviewer's suggestion, we have included the above figures in Supplementary Figure 5, supported by the following statement in the manuscript:

Line 448-460:

*"We identified no significant differences in B7/NP₃₀⁺ or B8/NP₃₀⁺ CD8⁺ T cell frequencies between HLA-B*07:02 and HLA-B*08:01 single-positive and double-positive individuals, however a trend for lower frequencies was observed in HLA-B*08:01-expressing donors (Supplementary Fig. 5a). A phenotypic comparison of B7/NP₃₀⁻ and B8/NP₃₀⁻ specific CD8⁺ T cell populations in HLA-B*07:02 and HLA-B*08:01 single-positive and double-positive individuals revealed a prominent B7/NP₃₀ T_{cm} population in HLA-B*07:02-expressing donors regardless of HLA-B*08:01 co-expression, while a visually apparent but statistically indistinguishable trend for higher B8/NP₃₀ T_{cm} frequencies was observed for HLA-B*08:01-expressing donors versus double-positive donors (Supplementary Fig. 5b). These results indicate that co-expression of HLA-B*07:02 and HLA-B*08:01 may result in lower recruitment of B8/NP₃₀⁺ CD8⁺ T cells, which could potentially be explained by the slightly reduced binding affinity of NP₃₀ to HLA-B*08:01 compared to HLA-B*07:02 (Supplementary Data 1)."*

Reviewer #2:

The manuscript in question is based on sound methods and interpretations and the issues raised below would not have a major impact on the conclusions drawn. Here is a summary of my understanding of the work performed:

Influenza B infections account for a significant fraction of seasonal influenza infections, yet are understudied as compared to Influenza A. The lack of known epitopes prevents the design of vaccines that target both IVA and IVB. The authors first established a ground truth by observing the immunogenicity of several known epitopes or HLA-B*08:01 in healthy individuals expressing that allele. Novel candidate epitopes were identified through an established immunopeptidomic pipeline. Peptides were stratified according to confidence in identification and predicted binding affinities and selected for further analysis. Peptide pools were tested for immunogenicity and further deconvoluted to determine the responses of individual peptides. HLA restrictions were confirmed through IFN γ and TNF α assays of pool-specific CD8⁺ T cells to peptide-pulsed C1R cells expressing the alleles of interest. The composition of the responding CD8⁺ T cells was determined through tetramer assays, with the majority of cells falling into the T_{cm} class. When stratified by age, it was determined that IBV-specific CD8⁺ T cell populations decline with increasing age. The frequency of IBV-specific CD8⁺ T cells was assessed and noted to be higher in lymph node and spleen than in PBMCs, although the highest frequency of the T_{rm} phenotype was in the lung. The crystal structure of the 11mer NS1 peptide in complex with HLA-B*07:02 was solved to reveal a bulge in the epitope conformation. The crystal structure of an NP epitope in complex with HLA-B*07:02 was obtained and extrapolated to HLA-B*08:01 and HLA-B*35:01 to explain the lack of responses detected against the latter. An analysis of TCR repertoires indicated relatively high diversity between individuals for each peptide:MHC as well as differential gene usage associated with each epitope. Motif analysis yielded divergent CDR3a and b motifs among the 4 peptide:MHC complexes studied, mainly driven by lower frequency clonotypes.

We thank the Reviewer for their remarks and the summary of our study.

Moderate concerns:

* In several places throughout the manuscript, the authors state that only 18 CD8+ IBV-specific epitopes have been reported in the literature. However, a search with these parameters at the IEDB (www.iedb.org) yields 39. Although this doesn't fundamentally change the outcome of the research, the significance may be overstated as a 50% increase in the number of known epitopes versus 20%. Can the authors help to clarify the discrepancy?

We thank the Reviewer for their thorough appraisal. The discrepancy between the IEDB search and the 18 previously identified epitopes in our manuscript is resulting from the fact that we have not counted peptide variants as separate epitopes (e.g. YFSPIR|TF and YFSPIRVTF) which are counted as individual epitopes by IEDB. We also only consider the minimum epitope and as such overlapping peptides are not counted as separate epitopes (e.g. KLGEFYNQM, KLGEFYNQMM and MVVKLGEFYNQMM). Importantly, we have also verified that IEDB reported epitopes were also functionally confirmed epitopes in the original studies. For example, some peptides listed in IEDB are not claimed to be epitopes by the papers themselves because they were not immunogenic in the majority of human donors used in the studies (e.g. IWVKTPLKL in HLA-A*24:02; Hensen et al. Nat. Commun. 2021). Taking this into account yields 18 known influenza B epitopes.

For clarity, we have added a detailed description of how previously discovered IBV epitopes were identified to our methods section:

Line 705-711-Methods: ***“Previously discovered IBV epitopes. The Immune Epitope Database (IEDB)²⁸ was used to search for known IBV CD8+ T cell epitopes. The parameters for the search were for linear peptides, human hosts, T cell assays and MHC Class I restriction. References for each peptide were analysed to ensure that epitopes were confirmed ex vivo, in vitro or in vivo, excluding peptides not claimed as being epitopes by the respective studies. Peptides with single amino acid substitutions or overlapping sequences were counted as a single epitope.”***

To further clarify our findings, we have also included the following footnote in Figure 1d:

“Overlapping peptides and peptide variants were counted as a single epitope (see Methods)”

* MHC:peptide binding affinity predictions were performed with NetMHC4.0, which was a state-of-the-art method in 2015, but has since been surpassed in performance by other methods, including NetMHCPan 3.0 and above. Would the outcome, in terms of peptides selected for further analysis, have changed if NetMHCPan 4.0 or higher was used for the predictions?

We thank the reviewer for their comment. In this study we used isolation of peptides directly from the HLA as a method to identify influenza epitopes that is not biased by the limitations of binding predictions. That said, we used NetMHC4.0 binding predictions predominantly to help differentiate the peptides that may be bound to the endogenous HLA of C1R (HLA-C*04:01 and HLA-B*35:03) as opposed to the transfected HLA allotype. We also used previously acquired data from C1R derivatives as noted in the materials and methods. Given the overexpression of the transfected HLA it is anticipated that most peptides identified will be isolated from the transfected HLA allotype.

Of the peptides we evaluated, 21/21, 32/35 and 42/53 MS-identified peptides tested for HLA-B*07:02, B*08:01 and B*35:01 respectively were predicted as binders using NetMHCpan4.1b EL % rank threshold of 0.5 and 2 for strong and weak binders respectively. Using these thresholds with NetMHCpan4.1b, for HLA-B*07:02 an additional 1 predicted weak binder, NP₁₆₅₋₁₇₃ (YFSPIRVTF), may

have been tested (Rebuttal Figure 5a), although it should be noted that this peptide is also predicted to bind HLA-C*04:01. For HLA-B*35:01, an additional 4 predicted binders may have been considered for testing (Rebuttal Figure 5b). HLA-B*08:01 peptides cover 34/49 of the in-frame binders predicted by NetMHCpan4.0 and 32/77 of the in-frame ligands predicted by NetMHCpan4.1 EL, indicating that our data may yet yield further HLA-B*08:01 restricted T cell epitopes (Rebuttal Figure 5c).

Rebuttal Figure 5 (reviewer only): Venn diagrams of the overlap in NetMHC4.0 predicted binders (% Rank ≤ 2 , green circle), NetMHCpan4.1b predicted ligands (% EL Rank ≤ 2 , blue circle) and tested peptides (orange circle) for HLA-B*07:02 (a), HLA-B*35:01 (b) and HLA-B*08:01 (c). This includes 8-14mer peptides only (for which binding predictions were performed). Note that for HLA-B*35:01 3 additional peptides were tested: PB₁₂₂₋₃₀ FPYTGVPY and NP₃₀₋₃₈ RPIIRPATL not detected in MS data, and 17mer PB₁₂₂₋₃₈ FPYTGVPYSHGTGTGY (binding predictions not performed).

* Methods, lines 739-745: Different thresholds, in terms of percentile rank and binding affinities, were applied to several alleles. In some cases, the thresholds seem interdependent. I could not find any justification for this approach in the manuscript as opposed to a uniform threshold across alleles.

We appreciate that the Reviewer for bringing the lack of clarity to our attention. The reported peptides were isolated from C1R cell lines overexpressing HLA-B*07:02, -B*08:01 or -B*35:01. However, C1R cell lines also express HLA-C*04:01 and low levels of HLA-B*35:03, and the possibility of binding these molecules was also considered. It is anticipated that the majority of peptides identified were derived from the overexpressed HLA of interest. Binding predictions and previously acquired data from B/Malaysia/2506/04 infected C1R derivatives were used to help delineate peptides bound to HLA-C*04:01 and HLA-B*35:03 rather than the overexpressed HLA of interest. Thresholds for strong and weak predicted binding were the same for all HLA. Due to their overexpression, for HLA-B*07:02, HLA-B*08:01 and HLA-B*35:01, an extra category of potential binders was included for peptides that were predicted to bind with a % rank less than 20, if there was

not a better explanation in binding of HLA-C*04:01 (or HLA-B*35:03 for HLA-B*08:01 only). Given the similar preference for Proline at P2 for HLA-B*35:03 restricted peptides as well as for -B*35:01 and -B*07:02 restricted peptides, there is a possibility that they bind similar peptides, hence binding prediction to HLA-B*35:03 was not considered to reduce the likelihood of binding to HLA-B*07:02/B*35:01.

We have now further clarified this in the methods, which now reads (Line 768-776):

*“Peptides were considered potential binders of HLA-B*08:01 if the % rank or nM Affinity was lower than that for B*35:03 and C*04:01 and the % rank below 20. For B*07:02 and B*35:01, peptides were considered potential binders if the % rank or nM Affinity was lower than that for C*04:01 and the % rank below 20. For B*07:02 and B*35:01 predicted binding to B*35:03 was not considered to reduce likelihood of binding the transfected HLA due to similar preference for Proline at P2.”*

* Data availability: I did not see an SRA identifier for the TCR sequencing data. Will this be made publicly available at SRA or another resource?

We thank the Reviewer for bringing this to our attention. We have now included the data availability paragraph in the Methods (Line 931-938), which reads:

“Data availability. TCR sequence data (Supplementary Table 5-8, Source data) has been deposited in Mendeley [DOI: 10.17632/wg7swwf6jr.1] and VDJdb [https://vdjdb.cdr3.net]. Mass spectrometry proteomics data have been deposited in the ProteomeXchange Consortium via the PRIDE⁸³ partner repository under the accession code PXD045000. Crystal structure have been deposited into the RSCB protein data bank (8TUH; HLA-B*07:02-NP₃₀₋₃₈ and 8TUB; HLA-B*07:02-NS1₁₉₆₋₂₀₆). Any additional information needed to reanalyze the data in this paper is available from the corresponding author upon reasonable request.”

In addition, we have now submitted the Nature Communications Reporting Summary form.

Minor concerns/questions:

* Table 1: The HA-255 epitope appears to be missing from this table.

We apologise for the oversight. The conservation of this peptide has been added to Table 1.

* Figure 6: The authors conclusion that IBV-specific T cell responses decline with age is mostly supported by the data presented in Figures 6b and c. However, the relationship appears to be clearer in the donors above 18 years of age. If the responses were averaged among the 3 groups, it looks as though the responses in the ‘children’ group would be similar to or lower than those in the ‘adults’ group in many cases. Have the data been grouped in this manner to evaluate whether the relatively low number of subjects in the children group are adding noise to the signal?

We thank the Reviewer for this observation. We have replotted the data in Figure 6b and c according to the Reviewer’s suggestion (please also refer to Rebuttal Figure 6). We agree with the Reviewer that, although not significant, the ‘children’ group tended to have lower responses compared to adults, which is in accordance with our recent findings for influenza A virus-specific CD8 T cells in children (van de Sandt 2023 Nature Immunology 4:1890-1907). The further increase in influenza-specific CD8⁺ T cell frequencies in adults may be the result of repeated influenza virus exposures boosting the respective T cell subsets. We have now rephrased the following parts of the results and discussion:

Results (Line 329-333): “Pooled analysis revealed that IBV-specific CD8⁺ T cell responses peaked in adults and declined in the elderly (Fig. 6b). Similar trends were observed within HLA-B*07:02⁺ donors and HLA-B*08:01⁺ donors but could not be confirmed in HLA-B*35:01⁺ individuals, due to the lack of available elderly HLA-B*35:01⁺ donors (Fig. 6c).

Discussion (Line 583-588): “We found that IBV-specific CD8⁺ T cell populations peaked in adults and declined in the elderly, similar to what we observed in IAV A2/M1₅₈-specific CD8⁺ T cells⁴⁰. However, in contrast to IAV-specific CD8⁺ T cells, the observed decline in IBV-specific CD8⁺ T cells was associated with reduced frequencies of naïve IBV-specific CD8⁺ T cells and further increasing T_{cm} populations.”

Rebuttal Fig. 6 (Fig. 6b and c in the modified version of the manuscript)

* The authors selected B/Malaysia/2506/04 as the representative strain, but did not provide rationale. It may be widely known in the field, but I'm not an IBV expert and wonder how this strain was selected.

B/Malaysia/2506/04 is a B/Victoria lineage virus which we have employed effectively in previous immunopeptidomics studies (Koutakos et al, Nat. Immunol. 2019; Hensen et al, Nat. Commun. 2021; Habel et al., Plos Pathog. 2022). Notably, B/Victoria is the only circulating IBV lineage after COVID-19 restrictions eased (Dhanasekaran et al, Nat. Commun. 2022) and thus was the only B lineage used.

* Conservation analysis: Are there more specifics on the query parameters used to download the IBV sequences from NCBI, or can you provide a list of identifiers? Was there any effort to remove redundancy and evaluate conservation among distinct sequences?

We thank the Reviewer for this suggestion. We have now provided all sequence accession numbers used for our analysis in Supplementary Data 2. The parameters used for our downloaded sequences are:

- Selected sequence type: Protein (amino acid sequence)
- Type: B
- Host: Human
- Country/region: Any region
- Protein: specific protein of interest (e.g. NP)
- Collection date: From 1940/01/01 to 9 September 2022 (HA), 26 November 2021 (NP) or 23 March 2023 (NS1).
- Full length only.

We did not collapse the sequences to ensure that we do not marginalize regional epidemic outbreak with a specific mutation.

* Figure 9d: I could not find information on the methods used for motif identification for the CDR3a and b regions. As GibbsCluster was mentioned in the epitope analysis, it is assumed that this tool was also used here. Can you clarify?

We apologize for the lack of clarity. GibbsCluster was used to assist immunopeptidome analysis. However, TCR derived CDR3 motif analysis was performed using TCRdist. Technical details on how TCRdist identifies motif can be found in Dash 2017 Nature. In brief, the program does not use GibbsCluster, but instead performs a statistical analysis of overrepresented CDR3 sequence motifs, which take in account the underlying sequence biases introduced by the rearrangement process. Furthermore, it uses a recursive search algorithm to identify sequence patterns that occur significantly more often in the observed receptors than in two V and J gene matched background sets of receptor sequences.

We have provided a description of this in the Methods (Lines 886-891):

“CDR3αβ motif analysis. TCRdist⁸⁸ was used to identify CDR3 motifs. In brief, TCRdist performs a statistical analysis of overrepresented CDR3 sequence motifs, and takes in account the underlying sequence biases introduced by the rearrangement process. Furthermore, TCRdist uses a recursive search algorithm to identify sequence patterns that occur significantly more often in the observed receptors than in two V and J gene matched background sets of receptor sequences.”

Regardless, is there a reason one might expect to identify a fixed-length motif, given the length variability of CDR3 sequences?

We apologise for the lack of clarity. The motifs are computed by the TCRdist algorithm and are reflective of CDR3 sequences of variable lengths.

We have added the above sentence to Methods (Lines 886-891).

* It might be worth including metrics regarding the conservation of the novel IBV epitopes among IAV strains to speculate about the potential for cross-protective responses.

We thank the Reviewer for their excellent suggestion. We have aligned the sequences of our

peptides to various human influenza A virus (IAV) strains starting from the early 20th century. Overall, these sequences of the newly identified IBV epitopes were not conserved in IAV (Rebuttal Table 1). As such, we added a sentence on IBV and IAV conservation to the manuscript (Line 290-291):

“Of note, our newly-identified IBV epitopes were not conserved among IAV strains (data not shown)”

Peptide	HLA	Influenza B	1918-1957 H1N1	1957-1968 H2N2	1968-2023 H3N2*	1977-2008 H1N1	2009-2023 H1N1
HA ₂₅₅₋₂₆₃	HLA-B*07:02	LPQSGRIVV	RD-A--MNY	N-ECD-LLS	AQSS---T-	FEAN-NLIA	QNADAYVF-
NP ₃₀₋₃₈	HLA-B*07:02 and HLA-B*08:01	RPIIRPATL	--NEN--HK	YSL---NIM	--NEN--HK	YSL---NEN	--NEN--HK
NS1 ₁₉₆₋₂₀₆	HLA-B*07:02	HPNGYKSLSTL	NE--RPP-TPK	DS-TVS-FQVD	NE--GPP-TPK	NET-GPPFTTT	DE--RPS-PPE
NP ₂₃₃₋₂₄₄	HLA-B*07:02	LPRRSGATGVAI	-----A-A-V	-----AGA-V	-----A-A-V	-----A-A-V	I--GK(L)STR--Q-
NP ₉₂₋₉₉	HLA-B*08:01	QMMVKAGL	IR-IKR-I	IR-IKR-I	IR---R-I	IR-IKR-I	R--ES-KP
NP ₄₇₉₋₄₈₇	HLA-B*08:01	QAVRRMLSM	EEI---CNI	EEI---CNI	AS-GK-IDG	AS-G--IGG	AS-G--IGG
HA ₂₃₁₋₂₃₉	HLA-B*35:01	SANGVTTHY	-HA-KSSF-	LG--CFEF-	MG--CFKI-	IG--CFEF-	IG--CFEF-
NS1 ₂₆₀₋₂₆₈	HLA-B*35:01	TAVGVLSQF	N-I--LIGG	AI--EI-PL	N-I--LIGG	N-I--LIGG	AI--EI-PL

Rebuttal Table 1: Parentheses indicates insertion. *Identical H3N2 sequences were collapsed as alignment was computationally intensive.

Reviewer #3:

The manuscript by Menon et al. explores influenza B (IBV) derived peptides presented by HLA Class I. The premise of the study is that IBV has not been well characterized for class I HLA restricted CD8+ T cell targets that potentially represent promising cellular vaccine candidates. The authors examine the global frequency of class I HLA alleles in their cohort and test the immunogenicity of IBV peptides in HLA-B*07:02, HLA-B*08:01, and HLA-B*35:01. Immunopeptidomics identifies 209 IBV peptides presented by HLA-B*07:02, HLA-B*08:01, and HLA-B*35:01. The authors condense these 209 peptides into seven pools for immunogenicity screening using PBMCs from healthy individuals with the aforementioned HLA-B alleles. The authors identify 9 novel IBV CD8+ T cell reactive peptides from NP, NS1, HA, and PB. Tetramer staining demonstrates TCM cells in the blood, TRM in the lungs, and that T cell phenotypes change with age. The presentation of NP30-38 by multiple HLA is then demonstrated with recognition by distinct TCR. The study then proceeds to structurally characterize NS1 196-206 and NP 30-38 in B*07:02. The stated impact is that considerably more IBV CD8+ T cell epitopes are now realized with a better understanding of CD8+ responses to these epitopes.

There are no apparent flaws or weaknesses with the experimental approaches. Indeed, the study is an experimental tour de force. The proteomics work is thorough, the immunogenicity testing that follows provides a complete hierarchy of T cell responsiveness/phenotype, and the tetramers facilitate the in-depth characterization of T cells restricted by these HLA/IBV complexes.

We thank the Reviewer for their encouraging remarks.

At this point, however, the study loses momentum – the reader anticipates a continued distillation of the HLA/IBV complexes and their corresponding T cell responses into a mechanistic determination of T cells to specific HLA/IBV complexes that either do or do not protect from IBV infection – the stated objective of the study. While the experiments to characterize the HLA/IBV complexes and T cells that recognize these complexes in different tissues and individuals are done with the utmost precision, the data never stratify into protective or non-protective HLA/IBV complexes and T cells that correspond to protective immunity; no correlates of immune protection to infection from IBV emerge from the data. No mechanism of HLA/IBV restricted CD8+ T cell immune protection from IBV infection emerges from the study. It is therefore difficult for the reader to ascertain the relevance of the data in this report as it pertains to influenza immunity.

We thank the Reviewer for these insights and apologise for the lack of clarity. The aim of our study was to identify and characterize novel IBV epitopes in healthy individuals and across the human lifespan and in tissues. To clarify this, we have rephrased the respective sentence in the introduction (Line 102-105) which now reads:

“The lack of known IBV epitopes not only hampers the development of universal IBV T cell vaccines, but also greatly impairs our ability to study protective versus detrimental IBV-specific immune responses across the human lifespan.”

We agree with the Reviewer that HLA/IBV restricted CD8+ T cell as correlates of immune protection is important but this was not within the scope of our study. However, it has been well established that CD8+ T cells provide protection against influenza A in humans (Sridhar et al., Nat. Med., 2013, Wang et al., Nat. Commun., 2015, Hayward et al., AJRCCM, 2015 and Forrest et al., CVI., 2008) and mice (McMaster et al., J. Immunol., 2015, Mytle et al., Viruses, 2021 and Zhao et al., mBio, 2018). Furthermore, previous work from our laboratory has also demonstrated the role of IBV-specific CD8+

T cells, identified through immunopeptidomics, in protection against influenza disease in HLA-expressing mice (Koutsakos et al., Nat. Immunol. 2019 and Hensen et al., Nat. Commun. 2021).

One study (Hayward et al., AJRCCM, 2015) utilised an IFN- γ enzyme-linked immunosorbent spot (ELISpot) on PBMCs stimulated with a pool of NP peptides and found that >20 spot-forming cells per 10^6 PBMCs was associated with reduced influenza virus nasal shedding. While Sridhar et al (2013 Nat Med) demonstrated that >50 spot forming cells (stimulated with overlapping peptide pools crossing multiple proteins) per 10^6 PBMCs prevented symptoms during the 2009 H1N1 pandemic. As a comparison, we calculated the frequency of tetramer⁺ cells per 10^6 PBMCs (Rebuttal Table 2) for three HLA-B7⁺/B8⁺ donors with whose samples were analysed for four tetramer⁺ populations. On average, these four tetramer⁺ populations sum to 22.2 tetramer⁺ cells per 10^6 PBMCs, similar to the 20 SFC per 10^6 PBMCs threshold defined by Hayward et al and approach those identified by Sridhar et al. Based on this and the literature we have cited, it is highly likely that the IBV-specific CD8⁺ T cell populations we identified here will contribute to protection against severe disease, particularly in adults where we identified the highest frequency of IBV-specific CD8⁺ T cells.

Donor	Tet ⁺ /10 ⁶ PBMCs					Total Tet ⁺ /10 ⁶ PBMCs
	B7/NP ₃₀	B7/NS1 ₁₉₆	B8/NP ₃₀	B8/NP ₉₂	B8/NP ₄₇₉	
A13	3.663462	12.6153846	3.733333333	3	0.857143	23.86932234
A16	25.71429	6.57142857	1.894736842	4	1.047619	39.22807018
A20	1.206897	0.68965517	0.375	1.142857	0.285714	3.700123153
					Average	22.26583856

Rebuttal Table 2

The latter part of the manuscript includes data pertaining to a peptide shared by both B*07:02 and B*08:01 that is 11 AA long and that bulges from the groove as demonstrated by crystallization. How is this relevant to the main tenet of the manuscript? These data are quite interesting and worthy of further exploration, but they do not fit the narrative of the manuscript.

We thank the Reviewer for this comment. The scope of our manuscript was to provide an in-depth analyses of CD8⁺ T cell responses directed against the newly-identified IBV epitopes. We contend that the structural work complements the functional and proteomics studies.

To expand, the epitope that is shared by B*07:02 and B*08:01 is NP₃₀₋₃₈, which is 9 aa long and has the correct anchor residues for HLA-B*35:01. We were interested to establish whether the CD8⁺ T cell responses directed against B7/NP₃₀ and B8/NP₃₀ were cross-reactive and understand why no responses were identified for B35/NP₃₀. Determining the structure of this peptide in complex with HLA-B*07:02 provides insight into the lack of B35/NP₃₀ specific T cell responses and distinct TCR repertoires of B7/NP₃₀⁺ and B8/NP₃₀⁺ CD8⁺ T cells, which corresponded to their lack of cross-reactivity. We have tried to further clarify the relevance by rephrasing the following two sentence in the result section, which now read:

Line 411-413: *“We sought to assess whether structural differences of NP₃₀₋₃₈ bound by HLA-B*07:02 or HLA-B*08:01 allomorphs could underpin similarities or differences in the responding TCR repertoires.”*

Line 432-437: *“Overall, the structural analyses indicated the NP₃₀₋₃₈ peptide can be accommodated with both HLA-B*07:02 and HLA-B*08:01. However, allele specific polymorphisms may result in differences of NP₃₀₋₃₈ presentation, depending on the allomorph. Our data also suggested that it was unlikely that*

*NP₃₀₋₃₈ would form a stable interaction with HLA-B*35:01, providing a potential explanation as to why no HLA-B*35:01/NP₃₀⁺ CD8⁺ T cell response was detected against this allomorph."*

The bulging of the 11mer NS1₁₉₆₋₂₀₆ peptide is relevant, as it may affect TCR engagement. Long peptides are known to pose a challenge for TCR recognition, especially if they display highly mobile conformations resulting in lower epitope-specific T cell frequencies (van de Sandt 2019 Nature Communications). Bulging peptides with stable conformations have been shown to elicit biased TCR repertoires (van de Sandt et al., Nat. Commun. 2019) and those with mobile conformations can elicit diverse repertoires (van de Sandt et al., Nat. Commun. 2019). Here, the NS1₁₉₆ peptide has a stable confirmation which may readily permit TCR recognition, and thus explain the prominence of B7/NS1₁₉₆⁺ CD8⁺ T cell responses and their diverse TCR repertoire. We have added the following statements to the respective section to further clarify its relevance:

Line 392-395: *"Long peptides can either have N or C terminal overhangs⁵⁰ or bulge out of the HLA-peptide binding cleft eliciting either a highly restricted⁵¹ or diverse TCR repertoire⁵² which can be associated with lower epitope-specific CD8⁺ T cell frequencies when associated with higher mobility³⁴."*

Line 404-407: *"Accordingly, the NS1₁₉₆₋₂₀₆ epitope forms a stable conformation that bulges out of the HLA-B*07:02 cleft, which is in accordance with the relatively robust B7/NS1₁₉₆-specific CD8⁺ T cell frequencies (Fig. 5b)."*

The authors make an argument in their discussion for their inclusion of a Yamagata strain-derived peptide in the study as being relevant due to the conserved nature of the peptide between it and the Victoria IBV strain. However, the language pertaining to a potential epidemic affecting millions due to waning nAb seems speculative.

We agree with the Reviewer and therefore removed the sentence from Discussion.

REVIEWERS' COMMENTS

Reviewer #1 (Remarks to the Author):

I am satisfied the responses and additional data provided.

Reviewer #2 (Remarks to the Author):

Thank you for the detailed responses and the additional clarifications added to the manuscript. All of the issues that I had raised have been thoroughly addressed.

Editorial note

This reviewer was additionally asked to comment in place of reviewer 3 who was unavailable to provide a comment during this round:

I believe that the authors have adequately addressed the concerns of Reviewer 3 by tactfully revising the stated goals of the manuscript, providing clearer motivations for and explanations of the structural work, and removing a speculative statement from the discussion.